# SAMReFiner: Taming Segment Anything Model for Universal Mask Refinement

**Yuqi Lin**[1,2], **Hengjia Li**[1], **Wenqi Shao**[2], **Zheng Yang**[3,*], **Jun Zhao**[3],
**Xiaofei He**[1,3], **Ping Luo**[2,4], **Kaipeng Zhang**[2,*]
[1]State Key Lab of CAD&CG, College of Computer Science, Zhejiang University
[2]Shanghai AI Laboratory   [3] FABU Inc.   [4] The University of Hong Kong
linyq5566@gmail.com   kp_zhang@foxmail.com

## Abstract

In this paper, we explore a principal way to enhance the quality of widely pre-existing coarse masks, enabling them to serve as reliable training data for segmentation models to reduce the annotation cost. In contrast to prior refinement techniques that are tailored to specific models or tasks in a close-world manner, we propose SAMRefiner, a universal and efficient approach by adapting SAM to the mask refinement task. The core technique of our model is the noise-tolerant prompting scheme. Specifically, we introduce a multi-prompt excavation strategy to mine diverse input prompts for SAM (*i.e*, distance-guided points, context-aware elastic bounding boxes, and Gaussian-style masks) from initial coarse masks. These prompts can collaborate with each other to mitigate the effect of defects in coarse masks. In particular, considering the difficulty of SAM to handle the multi-object case in semantic segmentation, we introduce a split-then-merge (STM) pipeline. Additionally, we extend our method to SAMRefiner++ by introducing an additional IoU adaption step to further boost the performance of the generic SAMRefiner on the target dataset. This step is self-boosted and requires no additional annotation. The proposed framework is versatile and can flexibly cooperate with existing segmentation methods. We evaluate our mask framework on a wide range of benchmarks under different settings, demonstrating better accuracy and efficiency. SAMRefiner holds significant potential to expedite the evolution of refinement tools. Our code is available at SAMRefiner.

## 1 Introduction

Image segmentation aims to assign a label to each pixel in an image such that pixels with the same label share certain characteristics. There are different notations about the group labels, such as semantic categories or instances. In the past few years, although significant progress has been made in image segmentation, the prevailing approaches rely on fully annotated training images, which are tedious to obtain. To reduce human labor, a labor-efficient alternative is generating segmentation masks by preceding models, especially those designed under incomplete supervisions (*e.g*, *unsupervised, weakly supervised or semi-supervised* annotations Wang et al. (2023b; 2022); Lin et al. (2023)). These generated segmentation masks can serve as pseudo labels to train advanced segmentation models or iteratively upgrade existing models Zhu et al. (2021); Yang et al. (2022). With the ever-increasing data amount, this pseudo-labeling paradigm showcases great practicality and potential to expand dataset volume for large-scale learning. However, the initial pseudo masks are usually noisy and lack fine details, particularly in object boundaries or in high-frequency regions (seeing Fig. 1a), hindering them from providing reliable supervision for model training.

Several mask refinement techniques have been proposed to improve the mask quality, but they suffer from major drawbacks: **1) model-dependent:** Some methods develop custom refinement modules tailored to specific networks and train them in an end-to-end fashion Zhang et al. (2021); Ke et al. (2022a), making them fail to work on different models. **2) task-specific:** Another group of techniques  Chen et al. (2022); Cheng et al. (2020); Shen et al. (2022) resort to model-agnostic

---

*Corresponding authors.

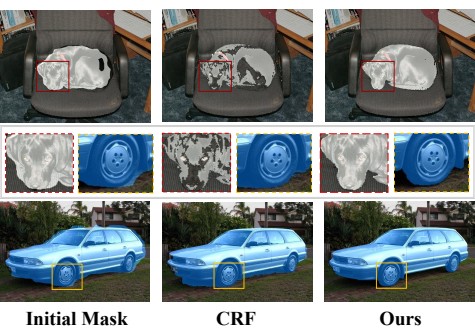
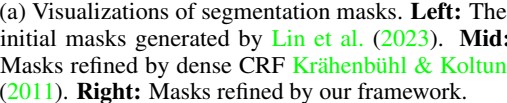

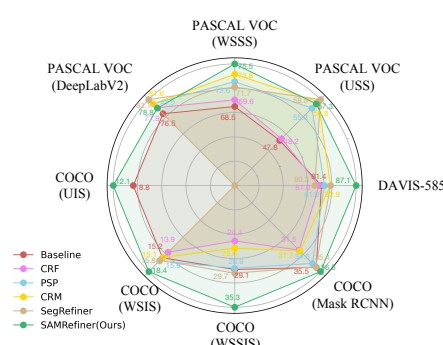

(a) Visualizations of segmentation masks. **Left:** The initial masks generated by Lin et al. (2023). **Mid:** Masks refined by dense CRF Krähenbühl & Koltun (2011). **Right:** Masks refined by our framework.

(b) The performance of our proposed mask refinement framework SAMRefiner on different benchmarks and comparisons with related works.

Figure 1: Visualization of segmentation masks and performance.

refinement mechanisms but they usually focus on specific task (*e.g*, semantic segmentation or instance segmentation). **3) category-limited:** Most previous works require training on target datasets with annotated data, limiting them to generalize to unseen categories and granularity. **4) time-inefficient:** Recent works Shen et al. (2022); Wang et al. (2023a) demonstrate better performance but refine one instance at a time, which is inefficient in complex instance segmentation tasks.

Recently, Segment Anything Model (SAM) Kirillov et al. (2023), an interactive image segmentation model that segments intended objects by user-provided prompts (*e.g*, point, box), has been proposed and achieved significant success in many image segmentation tasks. Some researchers have endeavored to adapt it to various tasks in order to take advantage of SAM's powerful representation capability to alleviate inadequate training samples. However, most of these studies focus on predicting masks from scratch and how to adapt SAM for the mask refinement task with pre-existing coarse masks remains an unexplored and challenging problem. We argue that this task is of great value in practical applications due to the widespread pre-existing masks (*e.g*, masks provided by offline models, inaccurate human annotations, or other forms of pre-processing). Making modifications on them could facilitate the annotation and benefit various downstream tasks.

However, since SAM is prompt-driven, adapting SAM to the refinement task is not trivial because it is difficult to obtain accurate prompts for SAM merely from coarse masks. Applying SAM directly to mask refinement using naive strategies would suffer from distorted prompts caused by noise and result in inferior performance. For example, in Fig. 2, we adopt the commonly used box prompt (tight box of coarse mask) and observe that this naive approach fails to obtain satisfactory performance because diverse types of errors (*e.g*, false-negative, false-positive) contained in the coarse mask would mislead the prompt extraction. Besides, results of directly taking the coarse mask as prompt are also terrible for SAM (the *4th* column in Fig. 2) due to its inherent nature in pre-training. (More prompt analyses are provided in the *Method Section* and *Appendix*.) ***Therefore, how to mine noise-tolerant prompts from the coarse mask poses a great challenge.***

In this paper, we tame SAM for the mask refinement tasks, which have unique characteristics compared to other segmentation tasks for the existence of coarse masks. We propose a universal and efficient framework called SAMRefiner, the core technique of which is the noise-tolerant prompting scheme. Specifically, to mitigate the effect of defects in coarse masks to prompt generation, we propose a multi-prompt excavation strategy to mine diverse and seemly prompts, including distance-guided points, context-aware elastic bounding boxes (CEBox), and Gaussian-style masks. These multi-prompts can collaborate with each other to generate high-quality masks and are more robust to noise than the single prompt. To overcome the confusion caused by multi-object cases, we introduce a split-then-merge (STM) pipeline to make it better suited for semantic segmentation. Meanwhile, given that the original SAM lacks dataset-specific priors, resulting in inaccurate IoU branch predictions, we propose SAMRefiner++. This approach incorporates an additional IoU adaptation step to enhance SAM's prediction accuracy on specific datasets by leveraging coarse mask priors. This minimal adaption startegy operates in a self-boosted manner and requires no extra annotations.

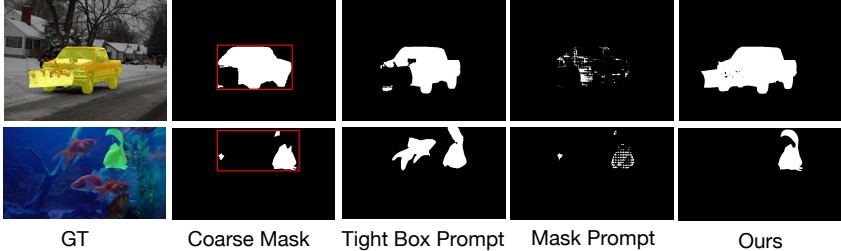

| GT | Coarse Mask | Tight Box Prompt | Mask Prompt | Ours |

Figure 2: Failure cases of SAM using the tight box of the coarse mask (red box) and directly using the coarse mask as the prompt. The tight box is sensitive to the false negative (first row) and false positive (last row) errors in the coarse mask, which would mislead SAM's predictions. And the separate mask prompt fails to work for SAM. Our proposed multi-prompt excavation strategy is robust to the noise.

We conduct experiments on a wide range of semantic and instance segmentation settings, with pseudo masks generated from incomplete supervisions, existing models, and synthetic data. Experimental results demonstrate the outstanding mask refinement capability of SAMRefiner (seeing Fig. 1b). Our approach is a generic post-processing tool and can be incorporated into any image segmentation approach in a self-training fashion with constant performance improvement.

Our contributions are summarized as follows:

- **New Roadmap:** SAMRefiner offers the first solution to address the mask refinement task based on SAM, which is of great value in practical applications.
- **New Method:** We uncover the deficiency of SAM in the mask refinement task and propose an effective and efficient framework to mine noise-tolerant prompts, successfully addressing the challenging universal mask refinement task.
- **Novel Insights:** While our work is based on SAM, it offers several novel insights and observations like the impact of mask prompt and the IoU adaption strategy.
- **Stronger practicality and performance:** This framework is versatile and can flexibly cooperate with existing segmentation methods under various setting. It significantly enhances the pseudo mask quality (*e.g*, over 10% for WSSIS) while taking less time (*e.g*, $5\times$ faster than CascadePSP).

## 2 RELATED WORKS

### 2.1 COARSE MASKS IN IMAGE SEGMENTATION

Coarse masks are common and ubiquitous in the image segmentation task due to their strict standard of pixel-accurate annotations. To relieve human burden, some works adopt the pseudo-labeling paradigm to obtain segmentation masks. These approaches usually leverage incomplete annotations (*e.g*, none, point, box, image-level labels or partially fully-labeled data) to obtain segmentation masks, which can be roughly categorized into *unsupervised* Cho et al. (2021); Ziegler & Asano (2022); Ke et al. (2022b); Hwang et al. (2019); Van Gansbeke et al. (2021); Zhou et al. (2022); Shin et al. (2022; 2023), *weakly-supervised* Lin et al. (2016); Dai et al. (2015); Papandreou et al. (2015); Ahn & Kwak (2018); Xie et al. (2022); Wang et al. (2020b); Xu et al. (2022b), and *semi-supervised* Wang et al. (2022); Filipiak et al. (2022); Yang et al. (2023b); Xu et al. (2022a). Although labor-efficient, the quality of pseudo mask is unsatisfactory, which can heavily impair the performance of subsequent segmentation model training. The noisy labels even exist in human annotation (*e.g*, MS COCO Lin et al. (2014)), which is inevitable for achieving pixel-accurate annotations at scale. This paper focuses on enhancing the quality of the coarse mask and consequently contributes to subsequent model training.

### 2.2 MASK REFINEMENT TECHNIQUE

To overcome the inaccuracy of coarse masks, several mask refinement methods have been explored Zhang et al. (2021); Kirillov et al. (2020); Xu et al. (2017); Zhang et al. (2019); Yuan et al.

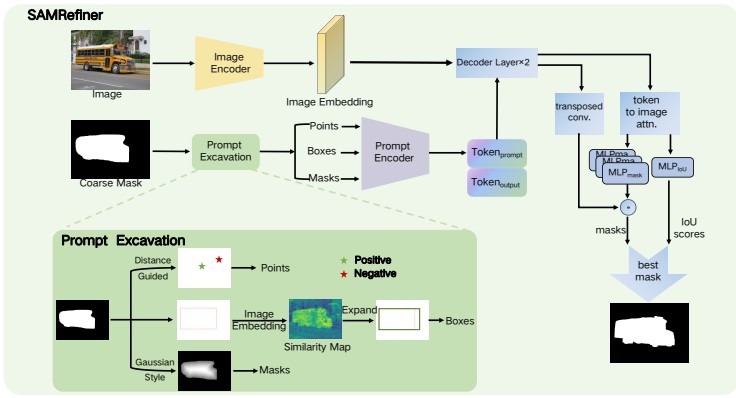
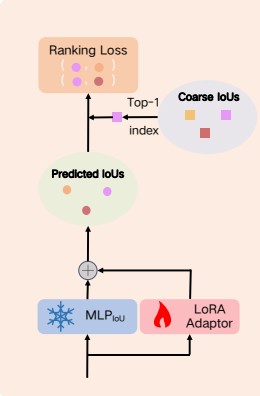

(a) SAMRefiner           (b) IoU Adaption

Figure 3: (a) An overview of our proposed framework. SAMRefiner leverages SAM to refine coarse masks by automatically generating prompts from coarse masks, including distance-guided points, context-aware elastic boxes and Gaussian-style masks. We select the best mask from multiple generated masks based on SAM's IoU predictions. (b) An overview of the introduced IoU adaption step, which aims to enhance the IoU prediction ability of SAM on specific datasets. We adopt a LoRA-style adaptor at the last layer of IoU MLP and a ranking loss is used to improve the top-1 accuracy of IoU predictions. This step is self-boosted and requires no additional annotation.

(2020). Most existing works are designed for specific networks or tasks and thus lack generality and flexibility. For example, PointRend Kirillov et al. (2020) and RefineMask Zhang et al. (2021) are built upon Mask RCNN He et al. (2017) for instance segmentation, BPR Tang et al. (2021) propose a model-agnostic post-processing mechanism but mainly focuses on instance segmentation. The dataset-dependant training in a close-world paradigm makes them overfit to specific datasets. CascadePSP Cheng et al. (2020) and CRM Shen et al. (2022) train on a large merged dataset and perform well across different semantic segmentation datasets, but the performance is poor on the complex instance segmentation setting. SegRefiner interprets segmentation refinement as a data generation process but the diffusion step is inefficient for practical use. Dense CRF Krähenbühl & Koltun (2011) is a training-free post-process approach but it lacks high-level semantic context and usually struggles to work in complex scenarios. Differently, we aim to design a versatile, generic and efficient post-processing tool across diverse segmentation models, tasks and datasets, which makes it a highly meaningful and valuable tool with broad applications.

## 2.3 SEGMENT ANYTHING MODEL

Segment Anything Model (SAM) has been considered as a milestone vision foundation model for promptable image segmentation. Several works have used this powerful foundation model to benefit downstream vision tasks, including object tracking Cheng et al. (2023); Yang et al. (2023a), image editing Gao et al. (2023), 3D object reconstruction Shen et al. (2023) and many real-world scenarios Ma et al. (2024); Han et al. (2023); Tang et al. (2023), while the potential of SAM in segmentation refinement task and the effect of different prompt types has been barely explored.

## 3 METHOD

In this section, we introduce our proposed mask refinement framework, as is shown in Fig. 3. We first review the architecture of SAM and its usage. Then, we introduce multi-prompt excavation strategies to exploit SAM. We further present an efficient adaption variant to enhance the accuracy of IoU predictions in a self-boosted manner.

## 3.1 REVIEW OF SAM

We start by introducing the components of SAM, which consists of an image encoder, a prompt encoder, and a mask decoder. 1) The image encoder is based on a standard Vision Transformer

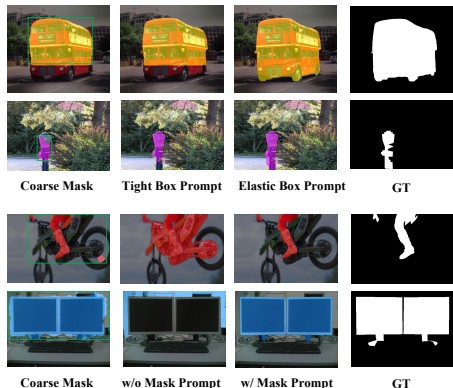

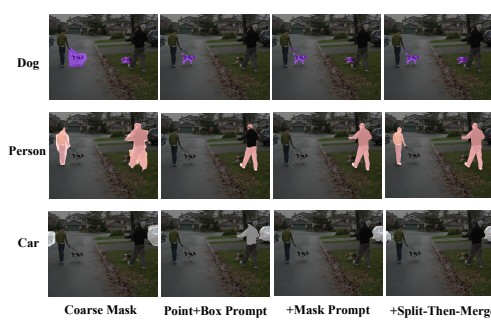

(a) Effects of the context-aware elastic box (top two rows) and mask prompt (last two rows).

(b) Effects of the proposed split-then-merge (STM) strategy.

Figure 4: Visualizations of our proposed techniques effects. All of them play a crucial role in mitigating the impact of defects in coarse masks.

(ViT) pre-trained by MAE He et al. (2022). It generates a $16\times$ downsampled embedding of the input image. 2) The prompt encoder can be either *sparse* (points, boxes, text) or *dense* (masks). For sparse prompts, points and boxes are represented as positional encodings summed with learned embeddings. Text prompts are processed by the text encoder of CLIP Radford et al. (2021). Dense prompts are directly convolved with the image embeddings and summed element-wise. 3) The mask decoder employs prompt-based self-attention and two-way cross-attention. This allows interaction between prompt-to-image and image-to-prompt embeddings, enabling simultaneous updates to the encoded image and prompt features. After two decoder layers, the output mask tokens are processed by a 3-layer MLP and then perform a spatially point-wise product with the upsampled image embedding to get target masks.

SAM is able to produce both a single mask or multiple masks (*i.e*, three masks) for each input prompt. The multi-mask mode is designed to address the ambiguity problem and an additional IoU token is adopted to learn the confidence of each mask, which reflects the IoU between each predicted mask and the target object. In Fig. 5a, we empirically find that the multi-mask mode is generally superior to the single-mask mode by simply selecting the mask with the best IoU predictions so we adopt the multi-mask mode in our experiments.

### 3.2 PROMPT EXCAVATION

As a promptable segmentation model, the input prompts play a crucial role in SAM because these prompts provide localization guidance of intended objects. To employ SAM for mask refinement, we need to mine prompts merely based on the initial coarse masks, which is challenging for the existence of noise and defects. Unlike previous works that mostly use one type of prompt Dai et al. (2023), our prompt excavation strategies aim to mine diverse and seemly prompts (including points, boxes and masks), making them collaborate with each other to mitigate the effect of defects in coarse masks. Note that SAM fails to work by merely using the mask as an input prompt, and we provide analysis in subsequent parts.

**Points.** The point prompt can provide position information for either foreground or background objects. However, it is difficult to determine the most salient point when using binary coarse masks. To solve this challenge, we leverage a simple but empirically effective object-centric prior: The center of an object tends to be positive and feature-discriminative, while uncertainty is mostly located along boundaries. Based on this criteria, we select the foreground point that has a maximum distance to the nearest background position as the positive prompt. Similarly, the negative prompt should satisfy the following principle: 1) the point is farthest away from the foreground region; 2) the point is within the bounding box of the foreground region.

**Boxes.** The box prompt shows a more powerful localization ability for the abundant cues it contains. Given a binary mask, it is simple to find the maximum bounding rectangle (tight box) of foreground

regions as the box prompt. However, the false-negative pixels in the coarse mask may hinder the quality of the bounding box, resulting in incomplete coverage of the potential object (Fig. 4a).

To address this, we propose a context-aware elastic box (CEBox) to adjust the tight box conditionally. The bounding box can be expanded in four directions according to the surrounding context. Specifically, the input image $\mathcal{I} \in R^{H \times W \times 3}$ is encoded as feature embedding $\mathcal{F}_{im} \in R^{h \times w \times c}$ in SAM latent space by the image encoder, where $(H, W)$, $(h, w)$ denote original image size and embedding size. The coarse mask $\mathcal{M}_{coarse} \in R^{H \times W}$ is resized to $\hat{\mathcal{M}} \in R^{h \times w}$ to keep aligned with $\mathcal{F}_{im}$. We calculate the mean feature embedding of the coarse mask (denoted as query embedding) as follows:

$$\mathcal{F}_{query} = \frac{1}{|\mathbb{1}_{\hat{\mathcal{M}}>0}|} \sum \mathbb{1}_{\hat{\mathcal{M}}>0}(\mathcal{F}_{im}) \tag{1}$$

where $\mathbb{1}_{\hat{\mathcal{M}}>0} \in \{0, 1\}$ is the indicator function to determine foreground regions, $|\cdot|$ represents the number of elements. We calculate the affinity between $\mathcal{F}_{query}$ and each spatial location in resized image embedding $\hat{\mathcal{F}}_{im} \in R^{H \times W \times c}$ to obtain a similarity map $Sim \in R^{H \times W}$ and binary it by 0.5:

$$Sim = [\mathcal{F}_{query} \cdot \hat{\mathcal{F}}_{im}]_{>=0.5} \tag{2}$$

For each direction in *{left, right, up, down}*, we enlarge the tight box $\mathcal{B}$ by 10% of the corresponding side length and approximate the positive ratio in the enlarged region $Sim_{context}$. A threshold $\lambda$ is used to determine the necessity to expand the current box in this direction. To avoid over-enlarge, we limit the maximum expanding pixels each time and run multiple iterations for progressive expansion.

**Masks.** Most existing works employ point or box as the initial prompt while mask prompt is usually discarded Dai et al. (2023); Chen et al. (2023); Zhang et al. (2023). The mask fails to serve as the initial input prompt for SAM separately (seeing qualitative results in Fig. 2 and quantitative results in Tab. 1). This is because the mask prompt merely acts as an auxiliary for point and box in the cascade refinement during SAM pre-training, with the predicted logits of the previous iteration as input to guide the next one. However, we argue that the mask prompt is vital to distinguish foreground and background in the mask refinement task, especially in the case that the box prompt fails to work (*e.g*, the oversized box results in falsely detected objects or background in Fig. 4a). Considering the inaccuracy of coarse mask, we leverage a Gaussian-style mask $GM$ based on distance transform used in point prompt:

$$GM(x, y) = \omega \cdot exp(-\frac{(x - x_0)^2 + (y - y_0)^2}{|\mathbb{1}_{\mathcal{M}_{coarse}>0}| \cdot \gamma}) \tag{3}$$

where $GM(x, y)$ represents the mask prompt at location $(x, y)$, $(x_0, y_0)$ is the *mask center point* that is farthest to the background regions, $\omega, \gamma$ are the factors to control the amplitude and span of the distribution. We provide a detailed analysis of the Gaussian Mask in the Appendix Appendix C.2.

**Application on Semantic Segmentation.** The semantic segmentation mask is category-wise and there may exist many objects in a semantic mask. In Fig. 4b, we find that SAM struggles to segment multiple objects with a large span (either miss-detect or falsely detect) using common prompts. Although the mask prompt can mitigate this problem, it fails when objects of different categories are mingled. We further propose a split-then-merge (STM) pipeline to solve it. **1) Split:** we split the mask by finding all connected regions. Note that some regions are noisy and trivial due to the inaccuracy of coarse masks. **2) Merge:** To form semantically meaningful regions, we iteratively merge the close regions based on the box area variation and mask area occupancy. Two regions will be merged only if the change of box area of (before and after region merging) is small and the mask area occupancy of the merged box is enough. An elaboration of this strategy is provided in Algorithm 1.

### 3.3 IoU Adaption

For SAM, the quality of the generated masks is determined by the input prompts, while the selection of the best mask is based on IoU predictions (denoted as $IoU_{pred}$). Traditional SAM leverages an individual token to produce the mask when multiple prompts are given (single-mask mode). However, in Fig. 5a, we empirically observe that selecting the best mask from multiple predictions of SAM based on $IoU_{pred}$ is generally superior to the single-mask mode under all prompt combinations. We give a detailed analysis in the Appendix Appendix C.4. However, in Fig. 5b, we find that

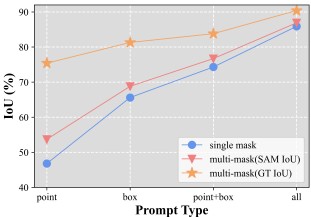 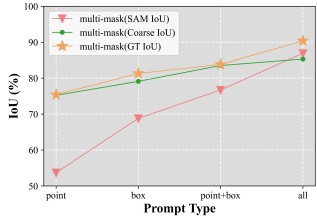 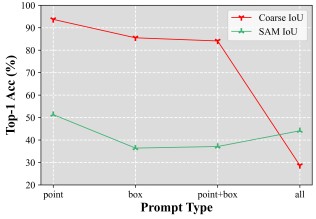

(a) Mask quality using different prompt types.

(b) The effect of different IoU selection criteria.

(c) Top-1 accuracy using different IoU selection criteria.

Figure 5: The effect of different prompt types, mask modes and IoU selection criteria on DAVIS-585.

mask selection based on $IoU_{pred}$ still falls short of the upper limit (select mask by ground-truth IoUs $IoU_{GT}$), which indicates the inaccuracy of SAM's top-1 IoU prediction. This is because the abovementioned SAMRefiner, which is training-free and generalized to most cases, is agnostic to downstream categories. SAM's IoU head is not specifically trained for intended objects, leading to suboptimal IoU predictions.

For the mask refinement task, where the GT masks are unavailable, we propose that coarse masks can act as effective priors to guide IoU predictions for domain-specific categories. To verify it, we denote the IoU between SAM's output mask and coarse mask as $IoU_{coarse}$, and compare the top-1 IoU accuracy of $IoU_{coarse}$ against $IoU_{pred}$. Results in Fig. 5c indicate that the top-1 performance based on $IoU_{coarse}$ outperforms $IoU_{pred}$ for simple point and box prompts, which is close to that based on ground truth. In contrast, the coarse IoU performs poorly in multi-prompt cases. It is likely that the less prompt provides ambiguous guidance for SAM and results in variant masks, enabling the coarse masks to provide effective guidance in selecting the intended one. However, the masks generated by multi-prompts have better quality than the coarse mask and thus may mislead the selection. To pursue better performance, we enhance SAM's IoU ranking ability by training under the single prompt case supervised by $IoU_{coarse}$ and expect it to benefit multi-prompt cases. **This process is conducted in a self-boosted manner and requires no extra annotations.**

Specifically, we focus on minimal adaptation of SAM toward better IoU predictions. To preserve the zero-shot transfer capability of SAM, we fix the model parameters of the pre-trained SAM and only add a LoRA-style adaptor Hu et al. (2021) in the IoU head, as is shown in Fig. 3b. Considering the inaccuracy of $IoU_{coarse}$, we adopt a ranking-based loss instead of a regression loss. In particular, for each SAM's predicted mask $M_i$ and its predicted IoU $x_i$, we calculate their coarse IoU and denote the index of the mask with the best coarse IoU as $j$. The pairwise ranking loss is computed as:

$$loss = \sum_{i=1, i \neq j}^{n} \max(0, x_i - x_j + m) \tag{4}$$

where $n$ is the number of total masks (3 for SAM), $m$ is the margin to control the minimal difference. This loss encourages the best IoU score $x_j$ to be higher than the remaining ones, thus promoting the accuracy of top-1 prediction. We train the adaptor based on the single prompt and use multi-prompt during inference. Note that despite LoRA's popularity, its optimal placement remains unclear. Previous works empirically place LoRA layers in some specific layers (*e.g*, backbone), altering existing knowledge to adapt to new domains, which modify the learned knowledge and affect the mask generation. In contrast, our approach inserts the LoRA layer in the IoU head, preserving SAM's full capability to generate high-quality masks while improving mask selection. To our knowledge, this minimal adaptation is underexplored and may provide new insights for the field. We denote SAMRefiner with this adaption step as SAMRefiner++, which only focuses on selecting better masks on the target dataset and has no effect on mask generation.

# 4 EXPERIMENT

## 4.1 EXPERIMENTAL SETUP

**Datasets and Implementation Details.** For a comprehensive evaluation of the mask refinement performance of SAMRefiner, we conduct experiments on a wide range of benchmarks, including those

Table 1: The quality of refined masks using different prompts and the effect of IoU adaption on DAVIS-585. Results are presented as SAMRefiner / SAMRefiner++

| Prompt Type | IoU | boundary IoU | Top-1 Acc |
|---|---|---|---|
| Coarse Mask | 81.4 | 71.4 | - |
| Point | 53.7 / 56.4 | 49.9 / 53.1 | 51.5 / 62.1 |
| Box | 68.8 / 70.8 | 61.9 / 63.5 | 36.4 / 56.8 |
| Mask | 37.3 / 40.4 | 32.6 / 33.5 | 28.4 / 30.8 |
| Point + Box | 76.7 / 79.1 | 69.0 / 70.9 | 37.1 / 53.7 |
| Point + Mask | 77.5 / 80.6 | 67.7 / 71.6 | 43.2 / 72.6 |
| Box + Mask | 84.6 / 85.1 | 74.2 / 75.4 | 36.2 / 60.7 |
| ALL | 86.9 / 87.1 | 75.1 / 75.4 | 44.1 / 63.8 |

designed for mask refinement (DAVIS-585 Chen et al. (2022)), instance segmentation (COCO Lin et al. (2014)), semantic segmentation (VOC Everingham et al. (2010)) under different settings. As a mask refiner, our method keeps the same setting as each baseline for pair comparison. The metrics we used include (boundary) IoU Cheng et al. (2021), (boundary) mask AP and mIoU. The threshold $\lambda$ and $\mu$ used in the box and mask prompt are set to 0.1 and 0.5 respectively. The factors $\omega, \gamma$ for Gaussian distribution are set to 15 and 4 by default. We adopt pre-trained SAM with the ViT-H image encoder as the segmentation foundation model and more details are provided in the Appendix.

## 4.2 ABLATION EXPERIMENTS

In this section, we conduct detailed ablation studies to analyze the effect of each component in our framework. We mainly experiment on DAVIS-585, as it is specifically designed for mask correction and contains various defects in the mask. We also leverage popular COCO and VOC to evaluate our method for specific scenarios.

**Effect of different prompts and IoU adaption.** Tab. 1 shows the performance of using different prompts for SAMRefiner. The results indicate that our proposed multi-prompt excavation strategy performs better than the single prompt. The mask prompt, which is barely considered in previous works, shows poor performance on its own but can bring an obvious advantage of nearly 20% IoU for point and box. Besides, we compare the mask selection by SAM's original IoU predictions (number before /) and our adapted IoU head (number after /). The IoU adaption step can significantly boost the top-1 accuracy of the best mask selection and further benefit the final IoU performance. We provide more ablation studies in Appendix for the page limit.

**Effect of different design choices in SAMRefiner.** We analyze the impact of different design choices for each prompt and report their relative contribution in Tab. 2. (1) We compare different strategies to sample the positive point prompt in Tab. 2a, including randomly choosing from the coarse mask, selecting the center of the bounding box, and selecting the point having a maximum distance to the background. Compared to random choice, using the box center shows even worse performance. This is because the bounding box is sensitive to the noise in the mask (*e.g*, the distant false positives), resulting in the inaccuracy of the box center. Our distance-guided are more robust to the noise and can obtain 52.5% IoU with only a positive point. The performance can be further improved to 53.7% by adding the extra negative point mentioned in Sec. 3.2. (2) Tab. 2b shows the impact of using the tight box and context-aware elastic box (CEBox). We adopt the coarse masks generated from PointWSSIS Kim et al. (2023) on COCO, which usually suffers from incomplete masks. The proposed CEBox can produce better boxes with higher $AP^{box}$ and benefit mask generation. (3) In Tab. 2c, we compare the commonly used mIoU with/without STM. Results show that STM can bring remarkable improvement for extremely coarse masks (*i.e*, 6.2% for MaskCLIP) and can constantly promote performance on better initial masks.

## 4.3 APPLICATION ON INCOMPLETE SUPERVISION

**Instance Segmentation.** To verify the effectiveness of our framework, we apply it to various typical methods, including unsupervised (CutLER Wang et al. (2023b)), semi-supervised (NoisyBound-

Table 2: Ablation study of our proposed strategies on different cases.

(a) Point sampling strategy.

| Point | IoU | bIoU |
|---|---|---|
| Random | 37.8 | 38.5 |
| Box Center | 22.2 | 26.3 |
| Distance-Guided | 53.7 | 56.4 |

(b) Context-aware elastic box.

| CEBox | $AP^{box}$ | $AP^{mask}$ | $AP^{boundary}$ |
|---|---|---|---|
| ✗ | 36.7 | 37.5 | 25.6 |
| ✓ | 38.2 | 37.8 | 25.9 |

(c) STM strategy.

| STM | MaskCLIP | CLIP-ES |
|---|---|---|
| ✗ | 51.1 | 79.1 |
| ✓ | 57.3 | 79.3 |

Table 3: Results of instance segmentation under different supervisions on COCO 2017. The Annotations denote the supervision type, including $\mathcal{U}$(unlabeled), $\mathcal{P}$(point-level label), $\mathcal{F}$(full labeled). Networks represent the final segmentation model trained based on the pseudo masks. We follow the default setting of each baseline method.

| Methods | Annotations | Networks | COCO train5K | | COCO val2017 | |
|---|---|---|---|---|---|---|
| | | | $AP^{mask}$ | $AP^{boundary}$ | $AP^{mask}$ | $AP^{boundary}$ |
| **Unsupervised** | | | | | | |
| CutLER | None | Cascade R-CNN | - | - | 8.8 | 2.8 |
| +SAMRefiner | None | Cascade R-CNN | - | - | 12.1(+3.3) | 5.0(+2.2) |
| **Semi-supervised** | | | | | | |
| NB | $\mathcal{F}$ 1% + $\mathcal{U}$ 99% | Mask R-CNN | 4.4 | 1.6 | 6.7 | 2.3 |
| +SAMRefiner | $\mathcal{F}$ 1% + $\mathcal{U}$ 99% | Mask R-CNN | 6.9(+2.5) | 4.4(+2.8) | 11.8(+5.1) | 6.5(+4.2) |
| NB | $\mathcal{F}$ 5% + $\mathcal{U}$ 95% | Mask R-CNN | 18.3 | 8.8 | 24.0 | 12.4 |
| +SAMRefiner | $\mathcal{F}$ 5% + $\mathcal{U}$ 95% | Mask R-CNN | 22.3(+4.0) | 14.4(+5.6) | 27.4(+3.4) | 16.5(+4.1) |
| NB | $\mathcal{F}$ 10% + $\mathcal{U}$ 90% | Mask R-CNN | 23.0 | 11.8 | 28.9 | 16.3 |
| +SAMRefiner | $\mathcal{F}$ 10% + $\mathcal{U}$ 90% | Mask R-CNN | 26.1(+3.1) | 17.0(+5.2) | 30.5(+1.6) | 18.6(+2.3) |
| **Weakly Semi-supervised** | | | | | | |
| PointWSSIS | $\mathcal{F}$ 1% + $\mathcal{P}$ 99% | SOLOv2 | 15.1 | 6.7 | 23.9 | 11.5 |
| +SAMRefiner | $\mathcal{F}$ 1% + $\mathcal{P}$ 99% | SOLOv2 | 25.4(+10.3) | 16.3(+9.6) | 30.2(+6.3) | 18.2(+6.7) |
| PointWSSIS | $\mathcal{F}$ 5% + $\mathcal{P}$ 95% | SOLOv2 | 32.3 | 19.7 | 33.4 | 19.6 |
| +SAMRefiner | $\mathcal{F}$ 5% + $\mathcal{P}$ 95% | SOLOv2 | 37.7(+5.4) | 25.9(+6.2) | 34.6(+1.2) | 21.6(+2.0) |
| PointWSSIS | $\mathcal{F}$ 10% + $\mathcal{P}$ 90% | SOLOv2 | 39.9 | 26.4 | 35.5 | 21.9 |
| +SAMRefiner | $\mathcal{F}$ 10% + $\mathcal{P}$ 90% | SOLOv2 | 42.8(+2.9) | 30.2(+3.8) | 36.1(+0.6) | 22.9(+1.0) |

ary Wang et al. (2022)) and weakly semi-supervised (PointWSSIS Kim et al. (2023)). Experiments are conducted on COCO following these methods. We evaluate the mask quality in terms of two aspects: 1) the performance of pseudo masks on the train set and 2) the performance of the final segmentation model trained based on these pseudo masks. The pseudo masks are evaluated on a subset of COCO train set (train 5K) and the final segmentation model Cai & Vasconcelos (2018); He et al. (2017); Wang et al. (2020a) is evaluated on the validation set. We compare both the commonly used mask AP and boundary AP. Results in Tab. 3 demonstrate the superiority of our framework. It can constantly boost the quality of pseudo masks in all settings, especially for those label-limited scenarios (*e.g*, the improvement for PointWSSIS with 1% annotations can reach 10.3%). Besides, the segmentation model can also benefit from refined masks, with a significant improvement under different settings, demonstrating that SAM can offer valuable knowledge and cues to improve these label-limited scenarios.

**Semantic Segmentation.** Tab. 4 shows the improvement of pseudo masks generated by unsupervised semantic segmentation (MaskCLIP Zhou et al. (2022)) and weakly supervised semantic segmentation (BECO Rong et al. (2023) and CLIP-ES Lin et al. (2023)). We refine the pseudo masks on the train set and use them to train a DeepLabV2 Chen et al. (2017) model following Lin et al. (2023). The results show that our method brings obvious performance gains for both the pseudo masks and segmentation models. The average improvement of pseudo masks is more than 5% and even approaches 10% for MaskCLIP and CLIP-ES. The superior performance across various datasets and settings demonstrates the generalization and flexibility of our framework.

## 4.4 COMPARISON WITH STATE-OF-THE-ART.

In Tab. 5, we compare our SAMRefiner with state-of-the-art model-agnostic refinement methods, including dense CRFKrähenbühl & Koltun (2011), CascadePSPCheng et al. (2020), CRMShen et al. (2022) and SegRefinerWang et al. (2023a). We first conduct experiments on previously used DAVIS-585, COCO and VOC. The results prove that 1) CRF shows inferior performance due to the lack of high-level semantic context and unfitness for the binary mask. 2) CascadePSP and CRM

Table 6: Performance of refined masks on COCO val set using LVIS annotations.

(a) Results on MaskRCNN.

| Method | $AP^{mask}$ | $AP^{boundary}$ |
|---|---|---|
| MRCNN(RN50) | 39.8 | 27.3 |
| +SegFix | 40.6 | 29.1 |
| +BRP | 41.0 | 30.4 |
| +SegRefiner | 41.9 | 32.6 |
| +SAMRefiner | 45.3(+5.5) | 35.9(+8.6) |
| MRCNN(RN101) | 41.6 | 29.0 |
| +SegFix | 42.2 | 30.6 |
| +BRP | 42.8 | 32.0 |
| +SegRefiner | 43.6 | 34.1 |
| +SAMRefiner | 46.6(+5.0) | 36.9(+7.9) |

(b) Results on more segmentation models.

| Method | $AP^{mask}$ | $AP^{boundary}$ | Method | $AP^{mask}$ | $AP^{boundary}$ |
|---|---|---|---|---|---|
| PointRend | 41.5 | 30.6 | SOLO | 37.4 | 24.7 |
| +SegRefiner | 42.8 | 33.7 | +SegRefiner | 40.5 | 31.3 |
| +SAMRefiner | 45.5(+4.0) | 36.0(+5.4) | +SAMRefiner | 44.1(+6.7) | 34.2(+9.5) |
| RefineMask | 41.2 | 30.5 | CondInst | 39.8 | 29.2 |
| +SegRefiner | 41.9 | 33.0 | +SegRefiner | 41.1 | 32.2 |
| +SAMRefiner | 44.7(+3.5) | 35.3(+4.8) | +SAMRefiner | 45.2(+5.4) | 35.8(+6.6) |
| MaskTransifiner | 42.2 | 31.6 | Mask2Former | 46.8 | 37.0 |
| +SegRefiner | 43.3 | 34.4 | +SegRefiner | 47.4 | 38.8 |
| +SAMRefiner | 46.3(+4.1) | 36.3(+4.7) | +SAMRefiner | 49.0(+2.2) | 39.0(+2.0) |

show competitive performance on semantic segmentation (VOC), but the improvement is limited or even worse than coarse masks on instance segmentation (DAVIS-585 and COCO). It is likely that these methods are trained on a merged dataset consisting of extremely accurate mask annotations, which has a strong relation to VOC and makes them fail to generalize to complex scenarios like COCO. We also explore the use of high-quality datasets on SAM (*i.e*, HQ-SAM Ke et al. (2023)) in the Appendix. 3) SegRefiner's performance is not stable across different settings because it lacks the ability to process diverse defects in the coarse masks. 4) SAMRefiner is more generic and can improve performance remarkably on various datasets due to its better robustness to the mask noise.

Table 4: Results of semantic segmentation under different supervisions on PASCAL VOC 2012. The Annotations denote the supervision type, including $\mathcal{U}$(unlabeled), $\mathcal{I}$(image-level label). Results on val set are based on training a DeepLabV2 model.

| Methods | Annotations | mIoU(train) | mIoU(val) |
|---|---|---|---|
| MaskCLIP | $\mathcal{U}$ | 47.8 | 47.3 |
| +*SAMRefiner* | $\mathcal{U}$ | 57.3 (+9.5) | 53.5(+6.2) |
| BECO | $\mathcal{I}$ | 66.3 | 69.5 |
| +*SAMRefiner* | $\mathcal{I}$ | 71.8(+5.5) | 70.9 (+1.4) |
| CLIP-ES | $\mathcal{I}$ | 70.8 | 70.3 |
| +*SAMRefiner* | $\mathcal{I}$ | 79.3(+8.5) | 74.9(+3.6) |

Table 5: Comparisons with SOTA methods. CM represents Coarse Mask.

| Source | CM | CRF | PSP | CRM | SR | Ours |
|---|---|---|---|---|---|---|
| **DAVIS-585** | | | | | | |
| DAVIS-585 | 81.4 | 81.0 | 81.9 | 82.9 | 80.3 | **87.1** |
| **COCO** | | | | | | |
| NB | 15.2 | 13.9 | 15.9 | 15.1 | 15.8 | **18.4** |
| PointWSSIS | 29.1 | 24.4 | 28.9 | 25.6 | 29.7 | **35.3** |
| MaskRCNN | 35.2 | 31.5 | 34.6 | 31.7 | 35.4 | **36.5** |
| **PASCAL VOC** | | | | | | |
| MaskCLIP | 47.8 | 48.2 | 55.3 | 56.8 | **58.5** | 57.3 |
| BECO | 66.3 | 66.5 | 68.4 | 69.0 | 68.7 | **71.8** |
| CLIP-ES | 70.8 | 72.6 | 76.9 | 78.7 | 74.7 | **79.3** |
| DeepLabV2 | 76.5 | 77.8 | 81.2 | 81.6 | **83.1** | 78.8 |
| Time (h) | - | 1.0 | 3.4 | 1.5 | 1.4 | **0.6** |

Besides, we compare the total time cost to refine masks for COCO train5K (with about 5K images and 37K masks). CRF is tested with 16 workers, and others are based on one 3090 GPU. SAMRefiner takes less than half the inference time compared to previous methods because SAM can batch process multiple masks in an image simultaneously, while other methods can only refine one mask each time. The batch processing capability makes SAMRefiner more efficient and competitive in practical use.

In addition, considering the original ground-truth annotations used in the COCO dataset are not accurate, we follow SegRefinerWang et al. (2023a) to evaluate the predictions of different fully supervised segmentation models on COCO val set using LVISGupta et al. (2019) annotations. Results in Tab. 6 indicate that our method outperforms other works by a large margin and can consistently enhance the mask quality generated by various networks (*e.g*, both CNN and Transformer), validating its generality for broad applications.

# 5 CONCLUSION

This paper uncovers the deficiency of SAM in the mask refinement task and proposes a universal and efficient framework called SAMRefiner to adapt SAM for mask refinement. We propose a multi-prompt excavation to generate diverse prompts that are robust to the defects in coarse masks. An optional IoU adaption step is introduced to further boost the performance on the target dataset without additional annotated data. We evaluate SAMRefiner on a wide range of image segmentation benchmarks under different settings, demonstrating its consistent accuracy and efficiency.

## 6 ACKNOWLEDGMENT

This work was supported in part by The National Key R&D Program of China (NO.2022ZD0160102).

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

---

**Algorithm 1** The Region Merging Strategy

---

**Input:**
    Region number $\mathcal{R}$; region label for each mask pixel $M_{label}$; a hyper-parameter $\mu$.
**Output:**
    Merged regions masks $\mathcal{M}^{stm}$.
1: Initialize $M_{merge} = M_{label}$,  $\mathcal{M}^{stm} = \varnothing$.
2: **for** $i = 1$ to $\mathcal{R}$ **do**
3:     $\mathcal{B}_i \leftarrow$ Extract minimum bounding box in $M_{label}^i$.
4:     $a_i^{box}, a_i^{mask} \leftarrow$ Compute areas for $\mathcal{B}_i, M_{label}^i$.
5:     **for** $j = i + 1$ to $\mathcal{R}$ **do**
6:         $\mathcal{B}_j \leftarrow$ Extract minimum bounding box in $M_{label}^j$.
7:         $a_j^{box}, a_j^{mask} \leftarrow$ Compute areas for $\mathcal{B}_j, M_{label}^j$.
8:         $(\bar{\mathcal{B}}, \bar{a}^{box}) \leftarrow$ Find merged boxes for $(\mathcal{B}_i, \mathcal{B}_j)$ and compute its area.
9:         **if** $(a_i^{box} + a_j^{box}) > \mu \cdot \bar{a}^{box}$ and $(a_i^{mask} + a_j^{mask}) > \mu \cdot \bar{a}^{box}$ **then**
10:             Merge region $i$ and region $j$.
11:             Update $M_{merge}$.
12:         **end if**
13:     **end for**
14: **end for**
15: $\mathcal{G} \leftarrow$ Extract merged region labels from $M_{merge}$.
16: **for** $k \in \mathcal{G}$ **do**
17:     $\mathcal{M}^{stm}$.append($M_{merge}^k$).
18: **end for**
19: **return** $\mathcal{M}^{stm}$.

---

# A   ADDITIONAL DETAILS

## A.1   DATASETS DETAILS

**DAVIS-585.** DAVIS-585 is proposed in FocalClick Chen et al. (2022) to evaluate the interactive mask correction task. It consists of 585 samples and generates the flawed initial masks by simulating the defects on ground-truth masks using super-pixels. There are different types of defects, *e.g*, boundary error, external false positive, and internal true negative, making it a comprehensive benchmark for the mask correction task.

**MS COCO 2017.** COCO comprises 80 object classes and one background class, with 118,287 training samples and 5,000 validation samples. We perform instance segmentation experiments on COCO following previous works Wang et al. (2022); Kim et al. (2023). To ensure a fair comparison, we maintain the same split of data subsets (*e.g*, 1%, 5%, 10%) as each baseline method. We assess pseudo labels quality by randomly sampling 5,000 images in the train set (denoted as train5K) that have no intersection with annotated data subsets.

**PASCAL VOC 2012.** We conduct semantic segmentation experiments on PASCAL VOC 2012 following Lin et al. (2023); Lee et al. (2021). It contains 20 categories and one background category. We evaluate the pseudo mask quality on the train set with 1464 images. An augmented set with 10,582 images Hariharan et al. (2011) is usually used for training in the WSSS task.

## A.2   IMPLEMENTATION DETAILS

We implement our method with PyTorch Paszke et al. (2019). For SAMRefiner, we didn't use multi-scale strategy and images are kept at their original sizes before being processed by SAM. For IoU adaption step, we use SGD optimizer with 0.01 learning rate. The batch size is set to 5 and we only train for 1 epoch. The learning rate is reduced to one-tenth at steps 60 and 100. We use margin ranking loss with the margin as 0.02 and the LoRA rank is set to 4. Note that the IoU adaption step is optional and we only adopt it on DAVIS-585. The time cost reported in the paper is tested on a single 3090 GPU. For instance segmentation, the threshold $\lambda$ is set to 0.1 for the box prompt. For semantic segmentation, $\mu$ used in the STM is set to 0.5. The factors $\omega, \gamma$ for Gaussian distribution are set to 15

Table 7: Quantitive comparison between automatic grid point prompt and our prompt strategy on DAVIS-585.

| Prompt Type | IoU | boundary IoU | Time (minute) |
|---|---|---|---|
| Coarse Mask | 81.4 | 71.4 | - |
| Max IoU | 70.6 | 65.5 | 8.0 |
| Merge | 81.9 | 73.1 | 8.0 |
| Ours | **86.9** | **75.1** | **1.6** |

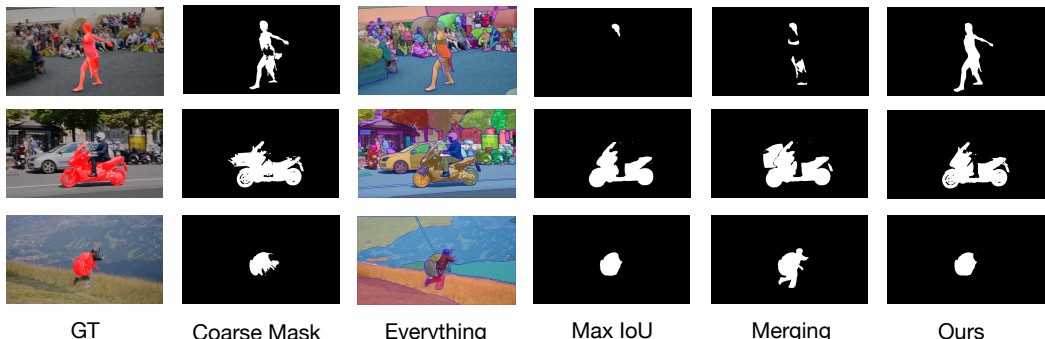

GT     Coarse Mask     Everything     Max IoU     Merging     Ours

Figure 6: Qualitive comparison with grid point prompts.

and 4 by default. We present the pseudo-code for the region merging strategy in Algorithm 1, which is an important component of our split-then-merge (STM) pipeline for semantic segmentation.

# B  ADDITIONAL EXPERIMENTS

## B.1  COMPARISON WITH AUTOMATIC MASK GENERATOR

SAM can produce masks for an entire image by sampling a grid of points over the image as prompts. This automatic manner can be used for mask refinement by matching the potential masks to the coarse mask. We validate its performance leveraging two matching criteria: 1) Max IoU: For each coarse mask, we select the SAM-generated segments with the highest IoU as the refined mask. 2) Merging: For each SAM-generated segment, it is viewed as a part of the final refined mask if the overlap area between this segment and coarse mask exceeds a certain percentage (*e.g*, 0.5) of this segment area Chen et al. (2023).

We compare our prompt excavation strategy with these two automatic grid-style point prompts in Tab. 7. We note that the performance drops severely for the Max IoU approach and barely improves for the Merging approach on DAVIS-585. It stems from the inherent drawbacks of this prompt generation manner, which is shown in Fig. 6. First, the grid point prompts split an image into several fine-grained masks and it is difficult to control the granularity. The best-matched mask selected by Max IoU usually fails to cover the whole object. Second, although the Merging strategy can obtain relatively complete objects, it is susceptible to defects in the coarse mask (*e.g*, false positives) and tends to result in over-detected. Thirdly, the SAM-generated segments are not exclusive and sometimes an object may be included in multiple masks with different granularity. It remains challenging to filter them out by the strategies above. In contrast to this bottom-up paradigm, our prompt excavation strategy directly produces diverse prompts for the target object (top-down paradigm), which is more purposive, accurate and robust to the noise in coarse masks. In addition, these grid-based prompts are inefficient (*i.e*, taking 5× more time than ours) because of the massive prompts and time-consuming post-processing (*e.g*, NMS Neubeck & Van Gool (2006)) to filter low-quality and duplicate masks.

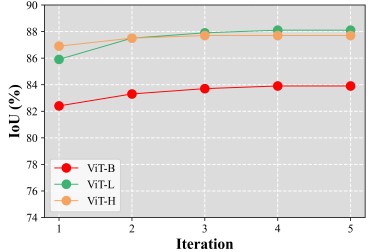 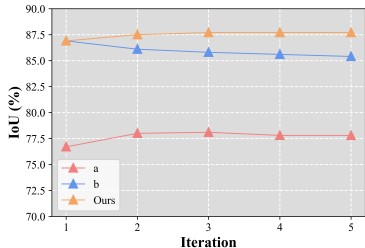

(a) Effects of different backbones and cascaded post-refinement.

(b) Comparison with SAM predicted mask logits.

Figure 7: Ablation study of different backbones and cascaded post-refinement strategies as well as mask logits.

## B.2 Effects of Different Backbones and Cascaded Post-refinement

The pre-trained SAM models are available with three backbone sizes and the mask can be iteratively processed by cascaded refinement. We compare the impact of using different backbones and iterations in Fig. 7a. The results show that the largest ViT-H outperforms other backbones at the first iteration, and multiple iterations can further improve the mask quality, especially for the ViT-L backbone.

Note that the mask prompt of each iteration in the cascaded refinement is from our prompt exvacation strategy (*i.e*, Gaussian-style mask). We also compare with some typical practices used in the original SAM, including **(a)** only using point and box prompt at the first iteration and adding SAM's predicted mask logits (produced by the previous iteration) at subsequent iterations; **(b)** using all prompts generated by our method at the first iteration but replacing the Gaussian-style mask with SAM predicted mask logits for subsequent iterations. Results in Fig. 7b demonstrate that SAM's mask logits can contribute to point and box (condition a) in the cascaded refinement but fail to work when our mask prompt is adopted in the initial step (condition b). This indicates that our Gaussian-style mask can provide more powerful guidance than the mask logits, which not only produce high-quality masks in the initial step but also more advantageous for cascaded refinement.

## B.3 Upgraded Results based on HQ-SAM

HQ-SAM Ke et al. (2023) is an advanced version of SAM that can enable more accurate segmentation. Our framework can also be applied to this powerful variant, and we conduct experiments on various benchmarks based on it to pursue better performance. In Tab. 8, we compare the performance of our framework using SAM and HQ-SAM on DAVIS-585, VOC (BECO Rong et al. (2023) and CLIP-ES Lin et al. (2023)) and COCO (NB Wang et al. (2022) and WSSIS Kim et al. (2023)). Results show that there is a significant improvement on DAVIS-585 and VOC while the performance is fair to SAM on COCO. This is because HQ-SAM enhances original SAM by specifically training on a high-quality dataset with large and salient objects, which aligns well with the characteristics of datasets like DAVIS-585 and VOC. In contrast, COCO has plenty of small objects and may not benefit as much from HQ-SAM.

Table 8: Comparison between SAM and HQ-SAM. We report IoU / boundary IoU on DAVIS-585, AP / boundary AP on COCO and mIoU on VOC.

| Model | DAVIS-585 | NB | PointWSSIS | BECO | CLIP-ES |
|---|---|---|---|---|---|
| SAM | 87.7 / 78.9 | 18.4 / 11.8 | 35.3 / 24.1 | 71.8 | 79.3 |
| HQ-SAM | 90.6 / 81.7 | 18.4 / 12.2 | 35.0 / 24.3 | 73.6 | 81.0 |

Table 9: Additional experiment results on BIG and relabeled PASCAL VOC datasets. The coarse masks are produced from DeepLabV3+ Chen et al. (2018).

(a) Results on BIG dataset.

| Method | Coarse Mask | SegFix | PSP | CRM | Ours |
|---|---|---|---|---|---|
| IoU | 89.4 | 90.0 | 92.2 | 91.8 | 93.9 |
| mBA | 60.2 | 69.3 | 74.6 | 75.0 | 74.8 |

(b) Results on relabeled VOC.

| Method | Coarse Mask | SegFix | PSP | CRM | Ours |
|---|---|---|---|---|---|
| IoU | 87.1 | 88.0 | 89.0 | 88.3 | 89.6 |
| mBA | 61.7 | 66.4 | 72.1 | 72.3 | 71.9 |

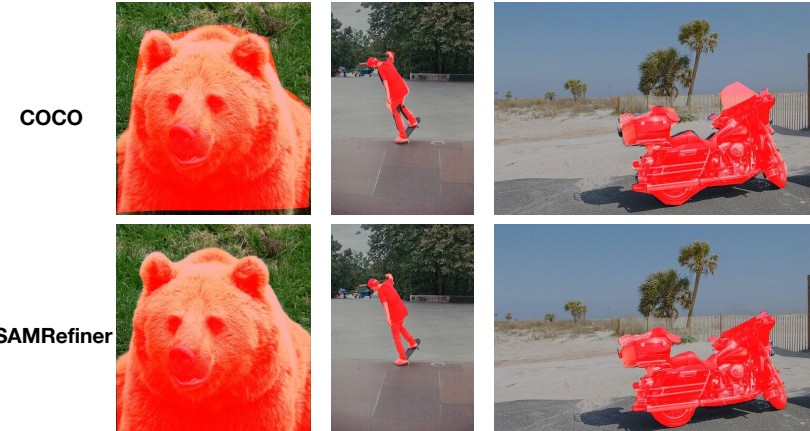

Figure 8: Visualizations of COCO annotations and our refined annotations.

### B.4 APPLICATIONS ON DIFFERENT TASKS

**Application on high-resolution images.** We evaluate our SAMRefiner on the BIG dataset, which includes ultra-high resolution images ranging from 2K to 6K. We directly refine the coarse masks generated by DeepLabV3+ Chen et al. (2018) based on SAM without dataset-specific finetuning, using IoU and mean Boundary Accuracy (mBA) as metrics following Cheng et al. (2020); Shen et al. (2022). Results in Tab. 9a show that our framework can effectively improve the quality of coarse masks and is superior or comparable to the previous methods by using the powerful HQ-SAM.

**Application on relabeled VOC.** CascadePSP Cheng et al. (2020) introduces a relabeled VOC dataset with accurate boundary annotations for better evaluation. We follow this setting to validate our framework on this benchmark. Results in Tab. 9b demonstrate the effectiveness of our method, with better IoU than existing methods. Note that the semantic obscurity may result in the inconsistency between human subjective annotations and SAM predictions, hindering SAM from obtaining better performance.

**Application on human annotations correction.** The human-annotated masks can also be coarse due to the strict standard of pixel-accurate annotations. For example, COCO Lin et al. (2014) masks are annotated in the polygon format, which is inaccurate in the boundary area (seeing Fig. 8). LVIS Gupta et al. (2019) constructs more precise annotations for COCO images. We refine the mask in the COCO val set using our SAMRefiner and evaluate them based on LVIS annotations. Results in Tab. 10 show that our methods can also work for inaccurate human annotations. There is a remarkable increase (*i.e*,

Table 10: Performance of refined masks on COCO2017 val.

| Data | $AP^{mask}$ | $AP^{boundary}$ |
|---|---|---|
| COCO | 38.3 | 27.3 |
| +Ours | 41.5(+3.2) | 33.0(+5.7) |

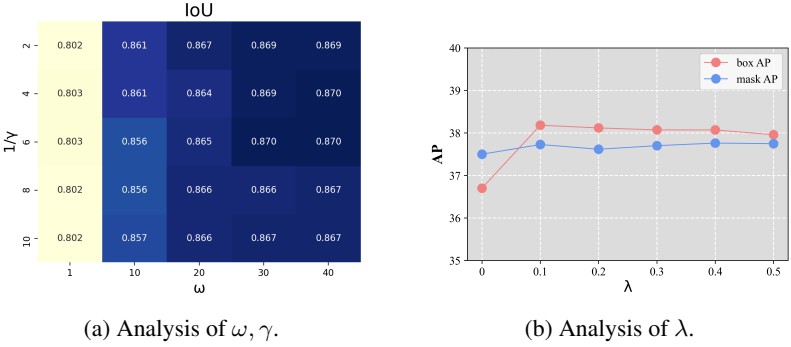

(a) Analysis of $\omega, \gamma$.          (b) Analysis of $\lambda$.

Figure 9: Ablation study of (a): $\omega, \gamma$ and (b): $\lambda$

3.2% mask AP and 5.7% boundary AP) for the mask quality. We provide qualitative comparisons in Fig. 8.

### B.5 ADDITIONAL ABLATION STUDIES

**Analysis of $\omega, \gamma$ in the mask prompt.** We leverage a Gaussian-style mask in our prompt excavation strategy, with two factors $\omega, \gamma$ controlling the amplitude and span of the distribution. We perform a sensitive analysis of these two parameters in Fig. 9a. When $\omega$ is too small (*i.e*, $\omega = 1$), the effect of mask prompts is negligible since the mask inputs of the original SAM are the predicted logits, which are not scaled to 0-1. We note that a relatively higher value for $\omega$ can promote mask prompts to benefit mask refinement, and the performance is not sensitive to these higher $\omega$ as well as $\gamma$.

**Analysis of $\lambda$ in the context-aware elastic box.** We introduce a threshold $\lambda$ in CEBox to determine whether to expand current boxes based on context features, which controls the trade-off of box sizes. We give an analysis of $\lambda$ in Fig. 9b. The proposed CEBox has consistent bonuses compared to the baseline ($\lambda = 0$) under different thresholds. We set $\lambda = 0.1$ in our experiments to avoid over-enlargement.

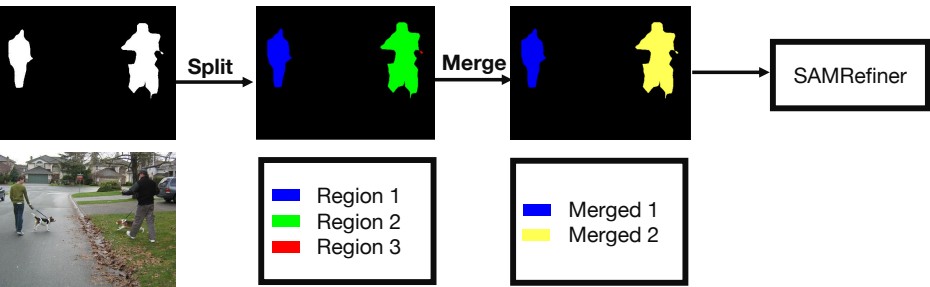

Figure 10: Illustrations of Split-Then-Merge (STM) pipeline. The *Region 3* in red color is small. Please zoom in for better visibility.

## C TECHNICAL DETAILS

### C.1 FURTHER ELABORATIONS OF STM

The Split-Then-Merge (STM) pipeline is proposed to solve the multi-object case in the semantic segmentation. In this case, SAM struggles to segment multiple objects with a large distance using common prompts, resulting in either missed detection or false detection (Fig. 4b). We propose STM to convert semantic masks with multiple objects into instance masks to ensure better compatibility with SAM. As shown in Fig. 10, STM includes two stages: 1) Split: split the mask by finding all

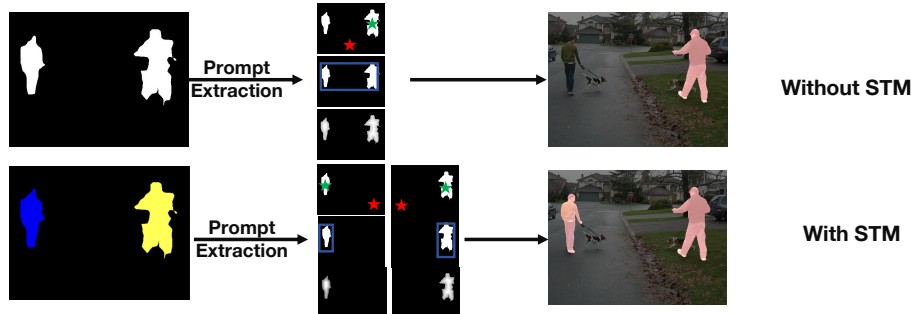

Figure 11: Visual comparisons between STM and baseline (without STM).

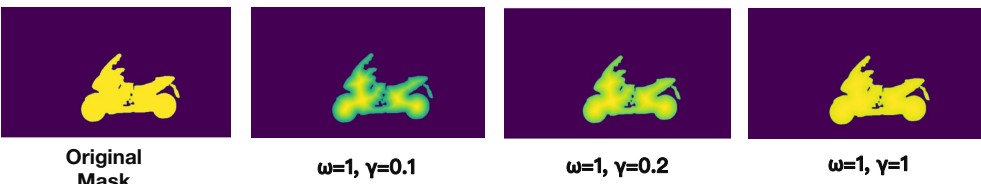

Figure 12: Visualization of Gaussian-style Masks under different $\gamma$.

connected regions, which tends to be messy and noisy; 2) Merge: iteratively merge the adjacent regions to form semantically meaningful regions based on the box area variation and mask area occupancy (Algorithm 1). The STM is performed before prompt extraction. Once finished, we can produce prompts based on the merged mask and leverage SAMRefiner for refinement. As shown in Fig. 11, STM can effectively mitigate the impact of multiple objects in semantic segmentation, yielding better results than the baseline.

### C.2 FURTHER ELABORATIONS OF GAUSSIAN-STYLE MASK

Note that the central point is not the geometry central point of the mask, but the farthest positive point selected by the previous point prompt step. We only apply the Gaussian operation to the foreground region of the mask, and the Gaussian-style mask is a generalized form of the coarse mask. For instance, when the amplitude $\omega$ is set to 1 and the span $\gamma$ is sufficiently large, the Gaussian-style mask is equivalent to the original coarse mask. Visualizations of the Gaussian Mask are presented in Fig. 12.

There are two main reasons for using the Gaussian-style mask: **1) Compatibility with SAM:** The original SAM doesn't support the binary masks as prompts. This is because the mask prompt merely acts as an auxiliary for point and box in the cascade refinement during SAM pre-training, with the predicted logits of the previous iteration as input to guide the next one. Therefore, the mask input for SAM requires logits with continuous values, while the original coarse mask is discrete-valued (0 and 1). The Gaussian operation can convert the binary mask to continuous, making it compatible with SAM. **2) The object-centric prior:** The center of an object tends to be positive and feature-discriminative, while uncertainty is mostly located along boundaries. The Gaussian-style mask effectively reduces the weights near boundaries. As shown in Fig. 9a, when $\omega = 1$, the performance drops significantly due to the incompatible value space, while the Gaussian transformed mask can consistently outperform the original coarse mask under different $\omega$ and $\gamma$.

### C.3 FURTHER ELABORATIONS OF CEBOX

For SAM, the image features of different instances (even within the same category) exhibit distinct characteristics. This enables SAM to produce fine-grained, component-level segments, making it support a variety of downstream applications. To illustrate this, we analyze feature similarity between different masks in Fig. 13a. As shown, the features of different instances, even within the same

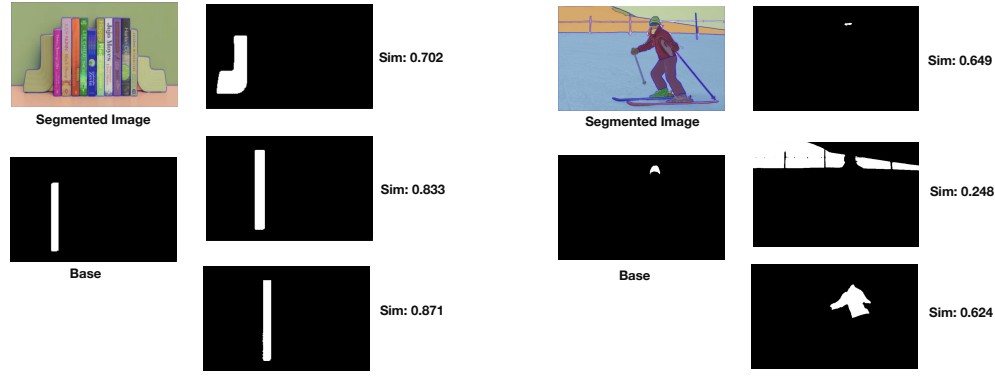

(a) Instance-level feature similarity.          (b) Component-level feature similarity.

Figure 13: Visualizations of feature similarity between base and other masks.

class, display certain differences. This characteristic allows SAM to distinguish between instances effectively (*e.g*, adjacent books). Similar conclusions can also be drawn for the part segmentation, as shown in Fig. 13b. On this basis, we can flexibly adjusting $\lambda$, a threshold to determine the necessity to expand the current box in each direction based on image feature similarity, according to different settings. For instance, a relaxed threshold could be applied for general segmentations, while a stricter threshold may be more suitable for fine-grained segmentations, such as distinguishing different instances or components.

## C.4    FURTHER ELABORATIONS OF IoU ADAPTION

Although the original SAM uses an individual token when multiple prompts are provided, we empirically observe that selecting the best mask from the remaining three masks based on the IoU prediction yields better performance than the fourth mask, as shown in Fig. 5a. This is because although the three predictions converge, some details remain different and usually better than the fourth token. We provide visualizations in Fig. 14 to compare the masks generated by different tokens. Though the improvement may not be remarkable, the advantage of IoU adaptation is that it doesn't require any additional annotated data and only takes advantage of the priors contained in the target dataset. SAMRefiner++ serves as a complementary enhancement to SAMRefiner when coarse masks on target datasets can provide high-quality guidance and this step is not mandatory.

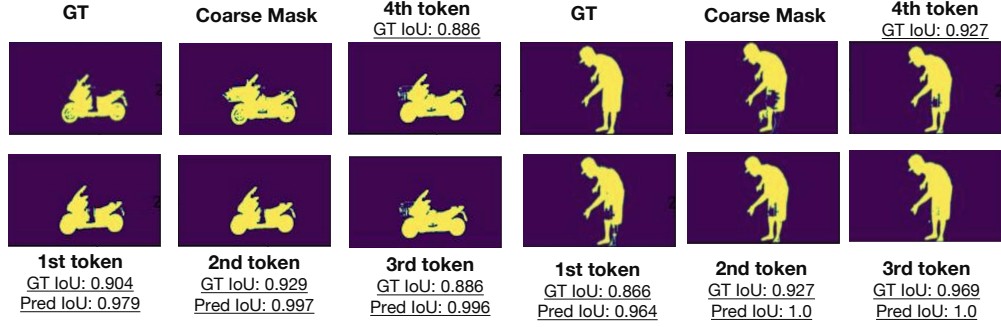

Figure 14: Visualization of masks generated by different tokens in SAM decoder.

## C.5    IMPACT OF THE DISTANCE-GUIDED POINT SAMPLING STRATEGY

The distance-guided point sampling strategy outperforms the box-center method as it effectively mitigates the impact of false-positive noise, which often distorts the bounding box and causes the box center to deviate from the actual object, as shown in Fig. 15.

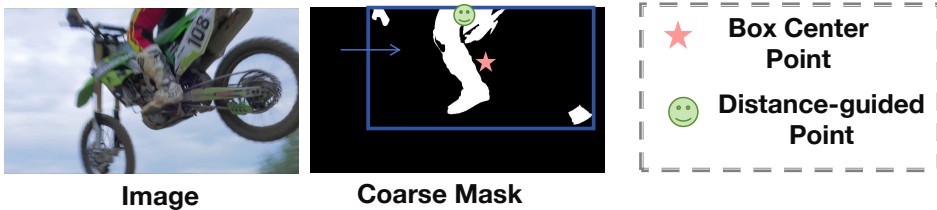

Figure 15: Comparison between box center point and distance-guided point.

### C.6 ANALYSIS OF THE QUALITY OF COARSE MASKS

In Fig. 16, we provide visualizations of the refined masks based on coarse masks with varying levels of quality. The results show that SAMRefiner works effectively when the coarse masks meet a certain quality standard but may fail when the coarse masks are extremely inaccurate. This is because the mask refinement task becomes an ill-posed problem if the initial mask is too coarse. For example, if the coarse mask only covers a person's head, reconstructing the entire person would be impossible without additional information due to the inherent ambiguity. Fortunately, most real-world coarse masks, such as those generated by model predictions, usually meet a certain quality standard and can be effectively handled by our proposed approach.

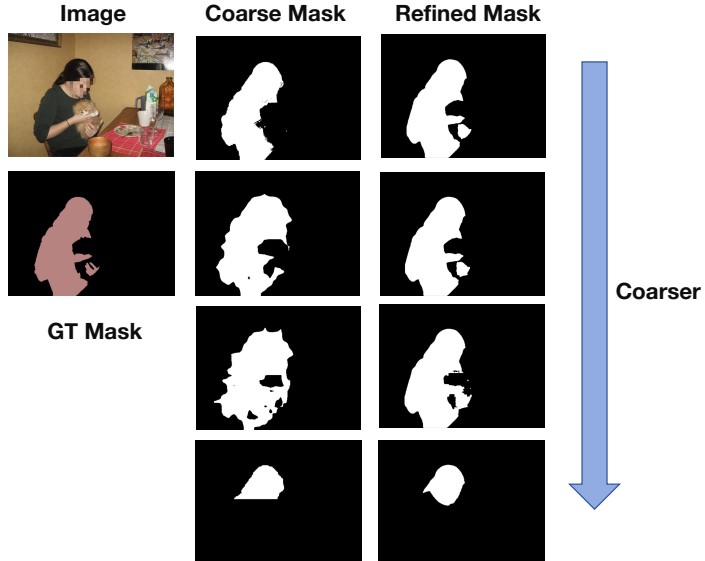

Figure 16: Visualizations of refined masks based on coarse masks with varying levels of quality.

## D DISCUSSIONS AND LIMITATIONS

### D.1 COMPARISON OF DIFFERENT MASK REFINEMENT METHODS.

We provide a detailed discussion of the differences between SAMRefiner and related methods (dense CRF, CascadePSP, CRM) in terms of the design principle, architecture, training data, advantages, and drawbacks in Tab. 11. Among these methods, dense CRF is a training-free post-process approach based on low-level color characteristics, making it efficient and easy to use. However, it struggles in complex scenarios due to its lack of high-level semantic context. CascadePSP and CRM, on the other hand, focus on aligning the feature map with the refinement target using CNN-based architectures. They are trained on a combined dataset with extremely accurate mask annotations and demonstrate

Table 11: Comparison of different mask refinement methods.

| Method | Design Principle | Architecture | Training Data | Advantages | Drawbacks |
|---|---|---|---|---|---|
| dense CRF | Maximize label agreement between pixels with similar low-level color | None | None | Training-free, Easy to use | inaccurate |
| CascadePSP | Align the feature map with the refinement target in a cascade fashion | CNN | MSRA-10K, DUT-OMRON, ECSSD, FSS-1000 | Class-Agnostic, Accurate on semantic segmentation | Task-dependent, Inefficient |
| CRM | Align the feature map with the refinement target continuously | CNN | MSRA-10K, DUT-OMRON, ECSSD, FSS-1000 | Class-Agnostic, Accurate on semantic segmentation | Task-dependent, Inefficient |
| SAMRefiner | Design noise-tolerance prompts to enable SAM for mask refinement | Transformer | SA-1B | Class-agnostic, Task-agnostic, Accurate, Efficient | Objects with intricate architecture |

strong performance on semantic segmentation tasks. Nevertheless, their performance on instance segmentation is less competitive, primarily due to the absence of complex cases in their training data and the inherent limitations of CNNs. Additionally, the cascade structure of CascadePSP and the multi-resolution inference required by CRM make them inefficient when handling masks with a large number of objects.

In contrast, SAMRefiner leverages the strengths of SAM by designing noise-tolerant prompts specifically for mask refinement tasks. This approach achieves better accuracy and efficiency compared to existing methods. Nevertheless, it may underperform for objects with intricate structures, a limitation inherited from SAM itself. This issue can be addressed using enhanced variants, such as HQ-SAM, as the experiments conducted in Appendix B.3.

### D.2 THE TARGET AND RELEVANCE OF SAMREFINER.

SAMRefiner is designed to be a universal framework for correcting coarse pseudo masks generated by various sources. This task is significant due to the wide source of coarse masks in practical scenarios, such as model predictions and even inaccurate human annotations. Our SAMRefiner can be treated as an effective post-processing method to improve data quality and the refined masks can then be used to train advanced models for specific tasks, which we denote as the pseudo-labeling paradigm. Although some recent works attempt to construct large foundation models to achieve open-vocabulary capability and show impressive performance, we argue that the pseudo-labeling paradigm still remains meaningful for certain application scenarios. First, people usually focus on limited semantic categories in specific practical use (*e.g*, automatic driving). The open-vocabulary setup is unnecessary and sometimes even detrimental to the performance of focused objects. Second, the foundation models tend to be large and inference-inefficient, which is not suitable for resource-limited and time-sensitive settings. Therefore, it is more efficient to customize specific models for different application scenarios instead of directly using large foundation models. Our framework treats the foundation model as a separate post-processor to improve the data quality of customized models, which is more generic and flexible. It can complement various segmentation methods and has the potential to be complementary to other refinement techniques and foundation models.

### D.3 APPLICATION SCENARIOS AND LIMITATIONS OF SAMREFINER(++).

**Application scenarios.** In this paper, we propose an effective mask refinement method SAMRefiner and its variant SAMRefiner++. SAMRefiner is a training-free method that refines masks using noise-tolerant prompts. It retains most of the characteristics of the original SAM and inherits its "universal capability." In contrast, SAMRefiner++ refer to the combination of SAMRefiner and IoU Adaption, which require additional training on target datasets. This method is specifically tailored for certain conditions and has strict prerequisites, such as the quality of coarse masks, which is dataset-dependent and may not achieve remarkable results on all datasets. As a result, SAMRefiner++ is not intended to be a universal method. Instead, it offers a potential approach to achieve further improvements without requiring additional annotations.

**Limitations.** For the mask refinement task, the final performance is highly affected by the quality of the initial coarse masks. The defects in masks are diverse, making it challenging to design a single effective method that applies to all scenarios. Our prompt excavation strategy proposes diverse prompt types to mitigate the effect of defects in the coarse masks. Although more noise-robust than

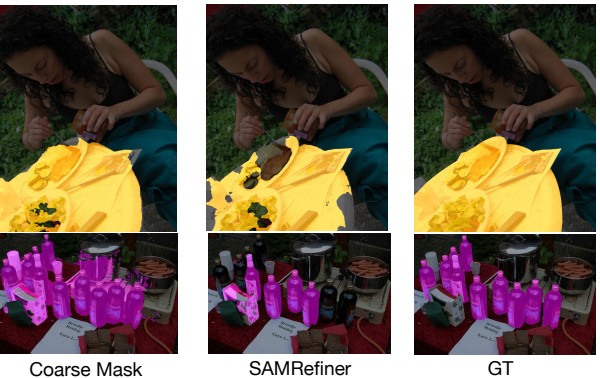

Figure 17: Failure cases on semantic segmentation.

previous works, it still fails to work when the initial masks are extremely noisy, as is shown in the last few rows in Fig. 18. Besides, there may exist semantic obscurity between SAM predictions and our subjectivity (*e.g*, whether the category *table* should contain the items on the table in Fig. 17), which is inevitable due to the lack of downstream data and a potential solution is dataset-specific adaption. Finally, SAM struggles to process multiple objects in semantic segmentation due to the absence of this condition during pre-training. Although our proposed STM can partly mitigate this, sometimes it fails to work (second row in Fig. 17). An option to consider is to finetune SAM to make it adapted to this setting.

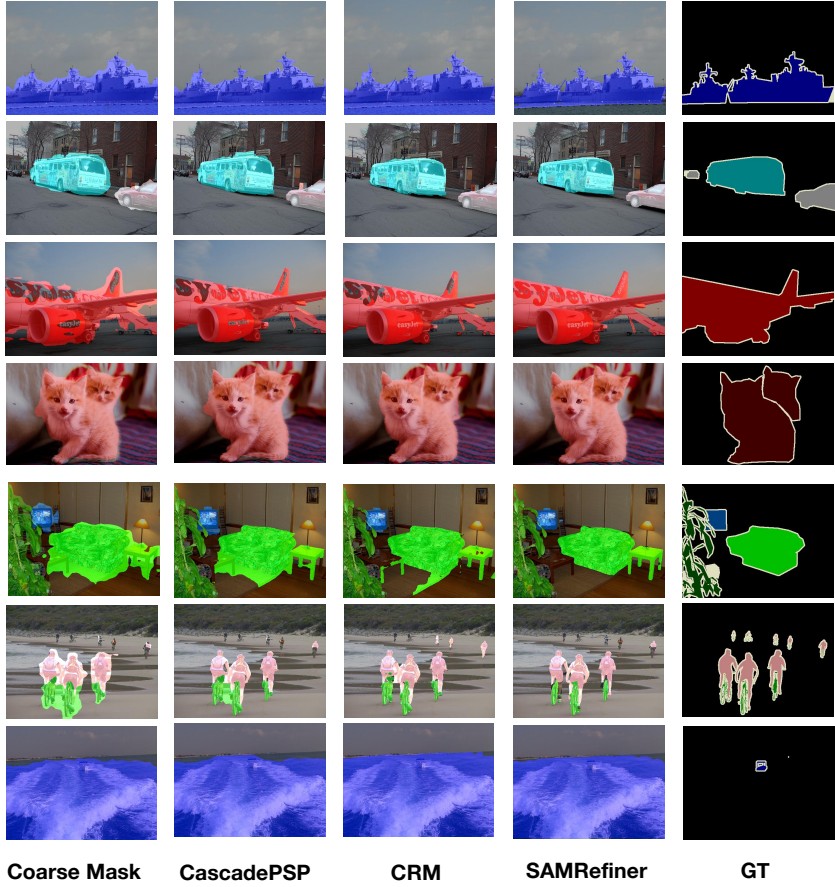

Figure 18: More visualizations on VOC. The last three rows show some failure cases.

# E    MORE QUALITATIVE RESULTS

In Fig. 18 and Fig. 19, we provide more qualitative results of our refined masks and previous works on PASCAL VOC 2012 and COCO 2017. We can observe that our SAMRefiner produces satisfactory segmentation results on boundaries and detailed structures. It is effective in both simple and complex scenes. The failure cases mainly stem from the extreme inaccuracy of coarse masks, resulting in false activation or missed detection.

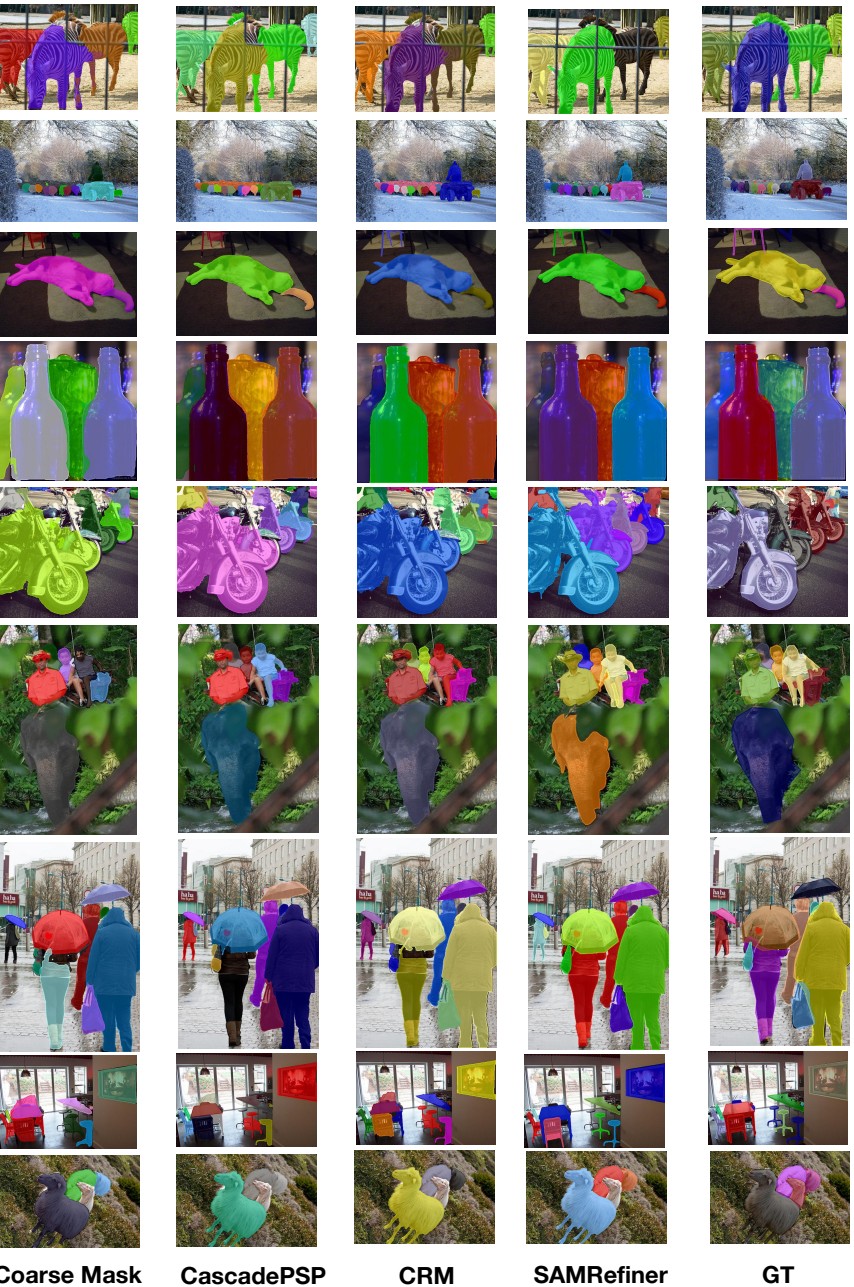

Figure 19: More visualizations on COCO. The last two rows show some failure cases.

