# OpenReview forum: "SAMRefiner: Taming Segment Anything Model for Universal Mask Refinement"
_ICLR.cc/2025/Conference — ICLR 2025 Poster_

### Official Review · Reviewer_ukvm · 2024-10-26

**Soundness:** 3
**Presentation:** 4
**Contribution:** 3
**Rating:** 6
**Confidence:** 4

**Summary:**

This paper proposes a new method to refine coarse segmentation masks by utilizing the Segment Anything Model (SAM). The method simultaneously leverages point-, box-, and mask-based prompts to produce high-quality refined masks. Additionally, an IoU-based adaptation method is proposed to improve the performance of the IoU head. Experiments on multiple tasks and datasets demonstrate the effectiveness of the proposed method.

**Strengths:**

1. This paper is well written and easy to follow.
2. The proposed method seems to be reasonable. The new designs of the point-, box-, and mask-based prompts used in this paper are novel and interesting.
3. The experiments are relatively comprehensive, demonstrating the effectiveness of the method across multiple tasks.

**Weaknesses:**

1. The performance of SAMRefiner++ is not reported in the experimental section, thus the effectiveness of the proposed IoU adaptation method cannot be evaluated.
2. It would be interesting if the authors could analyze the robustness of the method. For example, can the proposed method still maintain good performance when the initial mask becomes coarser?
3. As shown in Table 2(a), the proposed distance-guided point sampling method performs significantly better than the box center method. Therefore, it is suggested to provide a more detailed explanation of the advantages of the distance-guided method over the box center approach.
4. This method involves a lot of hyperparameters, such as λ, μ,  ω, and γ. It is suggested to include an ablation study on these hyperparameters for a more comprehensive evaluation.

**Questions:**

Please see the weakness section.

---

> ### Author Response · Authors · 2024-11-21
> **Response to Reviewer ukvm**
>
> ## Q1:
> > The performance of SAMRefiner++ is not reported in the experimental section, thus the effectiveness of the proposed IoU adaptation method cannot be evaluated.
>
> In Table 1, we reported the results in the form of  **SAM IoU / Adapted IoU**, where the number after "/" is the result of IoU adaption. Sorry to confuse you for our unclear expression. We have modified it to **SAMRefiner / SAMRefiner++** in the caption of Table 1 to enhance clarity.
>
> ---
>
> ## Q2:
> > It would be interesting if the authors could analyze the robustness of the method. For example, can the proposed method still maintain good performance when the initial mask becomes coarser?
>
> Thanks for your thoughtful suggestions. In the updated PDF **(Figure 17)**, we provide visualizations of the refined masks based on coarse masks with varying levels of quality. The results show that SAMRefiner works effectively when the coarse masks meet a certain quality standard but may fail when the coarse masks are extremely inaccurate. This is because the mask refinement task becomes an **ill-posed problem** if the initial mask is too coarse. For example, if the coarse mask only covers a person's head, reconstructing the entire person would be impossible without additional information due to the inherent ambiguity. Fortunately, most real-world coarse masks, such as those generated by model predictions, usually meet a certain quality standard and can be effectively handled by our proposed approach.
>
> ---
>
> ## Q3:
> > As shown in Table 2(a), the proposed distance-guided point sampling method performs significantly better than the box center method. Therefore, it is suggested to provide a more detailed explanation of the advantages of the distance-guided method over the box center approach.
>
> The distance-guided point sampling strategy outperforms the box-center method as it effectively mitigates the impact of false-positive noise, which often distorts the bounding box and causes the box center to deviate from the actual object. To illustrate this, we have provided a visualization comparing the box-centered and distance-guided point sampling strategies in the updated PDF **(Figure 18)**. We hope this vivid comparison aids in understanding the benefits of our approach.
>
> ---
>
> ## Q4:
> > This method involves a lot of hyperparameters, such as λ, μ, ω, and γ. It is suggested to include an ablation study on these hyperparameters for a more comprehensive evaluation.
>
> Your concern is exactly the results in our **Appendix (Section B.5)**. Please refer to this section for more details about hyperparameter analysis. We are happy to provide more details if the reviewer has any pieces of interest.

---

> > ### Comment · Reviewer_ukvm · 2024-11-21
> > **My concerns are adrressed**
> >
> > Thank the authors for the detailed responses. I think most of my concerns have been addressed, so I'm happy to keep my original score of acceptance.

---

> > > ### Author Response · Authors · 2024-11-24
> > >
> > > We deeply appreciate your acknowledgment of our work and the responses we provided. Best wishes to you.

---

### Official Review · Reviewer_a711 · 2024-10-31

**Soundness:** 3
**Presentation:** 3
**Contribution:** 3
**Rating:** 6
**Confidence:** 3

**Summary:**

The paper introduces SAMRefiner for refining coarse masks using the SAM. SAMRefiner utilizes a prompting scheme that generates varied prompts to address defects in initial masks, making it robust across different segmentation tasks. The paper also presents an extended version to add an IoU adaptation step to improve performance without requiring extra annotations. Experiments show that the proposed method enhances mask accuracy and efficiency, offering potential for reducing annotation efforts in training segmentation models.

**Strengths:**

1. This work introduces a novel method to enhance downstream segmentation models using SAM.
2. The exploration of prompt design in SAM is thorough, with comprehensive ablation studies provided to validate the design choices.
3. The IoU adaptation is interesting and provides new insights into SAM’s IoU prediction.
4. Overall, the paper is well-written, with elaboration on the prompt design.

**Weaknesses:**

1. Although this paper presents technical novelty, it needs to address a fundamental issue. For instance, the comparisons in Tables 3 and 6 involve SAMRefiner using SAM trained on a larger dataset, which introduces fairness concerns. To address this, the core of the paper should focus on how to better utilize SAM to enhance segmentation models. Indeed, the use of SAM is not limited to the refinement of Coarse Masks to indirectly improve downstream segmentation models. Alternative methods, such as straightforward distillation or the use of pre-trained weights, could also be explored. The paper should demonstrate the advantages of the proposed method in utilizing SAM from perspectives of training costs and downstream model performance, to justify its practicality and address fairness concerns.

2. The description of the IoU Adaptation section is somewhat confusing. It would benefit from using more precise symbols to differentiate between different types of IoU and to clearly describe the corresponding training strategy.

**Questions:**

Based on the concerns raised in the weaknesses section, please explain in detail the direct advantages of using SAMRefiner to enhance downstream segmentation model performance compared to the simple use of SAM, such as through distillation or pre-trained weights.

---

> ### Author Response · Authors · 2024-11-21
> **Response to Reviewer a711**
>
> ## Q1:
> > Although this paper presents technical novelty, it needs to address a fundamental issue. For instance, the comparisons in Tables 3 and 6 involve SAMRefiner using SAM trained on a larger dataset, which introduces fairness concerns. To address this, the core of the paper should focus on how to better utilize SAM to enhance segmentation models. Indeed, the use of SAM is not limited to the refinement of Coarse Masks to indirectly improve downstream segmentation models. Alternative methods, such as straightforward distillation or the use of pre-trained weights, could also be explored. The paper should demonstrate the advantages of the proposed method in utilizing SAM from perspectives of training costs and downstream model performance, to justify its practicality and address fairness concerns.
>
> Thanks for your insightful advice on how to better utilize SAM to enhance the segmentation model. Compared to directly using pre-trained weights, taming SAM for mask refinement offers several distinct advantages:
>
> **1)  Flexibility:** Mask refinement decouples SAM from downstream segmentation training, providing greater flexibility and compatibility across different tasks.
>
> **2) Efficiency and Adaptability:** The use of distillation or pre-trained weights suffers from additional training cost or limited architecture, while refined masks can directly replace ground-truth masks, making them more adaptable to downstream tasks.
>
> **3) Limitations of Pre-trained Weights with Coarse Masks:** Pre-trained weights often struggle to achieve strong performance when relying solely on coarse masks. For instance, when we use SAM’s pre-trained image encoder as the backbone (ViT-B + DeepLabV3) for semantic segmentation on VOC with coarse masks from CLIP-ES, the results **in Table R4** highlight significant limitations. This is because, although pre-trained weights provide a strong initialization, coarse masks fail to offer adequate supervision during fine-tuning. This highlights the need for tailored strategies to address noisy supervision, which could be an interesting future direction. By contrast, our method maximizes SAM's potential through the use of effective prompts, which may offer potential insights for the use of pre-trained weights.
>
> **4) Complementarity:** Our method **is orthogonal to** methods based on distillation or pre-trained weights and can be combined with them to achieve further performance improvements (e.g., from 63.9% to 71.0% in **Table R4**).
>
> **Table R4: Comparison of using SAM’s pre-trained weights and mask refinement.**
>
> |Model|Train Data|mIoU|
> |---|---|---|
> |SAM-ViT-B+DeepLabv3|Coarse Masks|63.9|
> |SAM-ViT-B+DeepLabv3|Refined Masks|71.0|
> |SAM-ViT-B+DeepLabv3|GT Masks|75.5|
>
> ---
>
>
> ## Q2:
> > The description of the IoU Adaptation section is somewhat confusing. It would benefit from using more precise symbols to differentiate between different types of IoU and to clearly describe the corresponding training strategy.
>
> We sincerely apologize for the confusion caused by our earlier unclear explanations. The IoU Adaptation (**Section 3.3**) has been revised in the PDF (**highlighted in blue**). We hope this revision makes our method easier to understand.

---

> > ### Comment · Reviewer_a711 · 2024-11-25
> >
> > Thank you for the detailed responses and revisions that address my concerns, so I will maintain the initial score favoring acceptance.

---

> > > ### Author Response · Authors · 2024-11-25
> > >
> > > We deeply appreciate your acknowledgment of our work and the responses we provided. Best wishes to you.

---

### Official Review · Reviewer_XLPd · 2024-11-02

**Soundness:** 2
**Presentation:** 3
**Contribution:** 2
**Rating:** 6
**Confidence:** 5

**Summary:**

This paper aims to adapt the segment anything model for the mask refinement task. It aims to go beyond the previous mask refinement networks that are 1) model-dependent, 2) task-specific, and 3) category-limited. It also aims to improve the time efficiency of the existing method. The key of the proposed method is prompting SAM through prompt excavation design and IoU adaptation.  To this end, the proposed method has been validated over DAVIS-585 (mask refinement), COCO (instance segmentation), and VOC (semantic segmentation).

**Strengths:**

Given that inaccurate pseudo-labels and coarse human annotations are widely used to train the deep segmentation model, mask refinement is a meaningful task to improve the training data and inference performance. Specialized deep learning mask refinement methods and non-learning-based methods have been proposed previously. However, they either fail to generalize to more diverse tasks or are limited in being adaptive to different samples. The proposed method, SAMRefiner,  aims to adapt SAM, a segmentation foundation model for different masks, which is of practical meaning.

The key contribution of the proposed method is to design noise-tolerant prompts based on the coarse mask. Since SAM is trained in a specific way of prompt combination. The author designed several strategies to adapt the SAM for the mask refinement task by combining point, box, and mask prompt.

Besides, IoU adaptation is used to adapt SAMRefiner for tasks and dataset-specific applications.

**Weaknesses:**

I have concerns about the novelty and the experiment:

1. Point Prompts Sampling Method: The distance-based sampling method used is not original, as it is a widely adopted approach for evaluating interactive segmentation models. The author suggests sampling one positive and one negative point, but there is no ablation study to support this design choice (e.g. why not other numbers). Different samples or tasks may have varying characteristics, so the author should provide analysis or empirical ablation to justify this approach.

2. Box Prompt Generation Method: Similar concerns apply to the box prompt generation method. A simple baseline would be to take the bounding box over the coarse masks. The appropriateness of the proposed method heavily depends on the quality and characteristics of the coarse mask. While the author provides an ablation study using coarse masks generated by PointWSSIS, this is specific to the dataset and method. The author claims the proposed method is general, but additional analysis or empirical observations are needed to support this claim. The authors are suggested to evaluate the box prompt generation method on a wider range of datasets and coarse mask types to demonstrate its generality. Additionally, proposing a comparison against simpler baselines like the tight bounding box across various datasets would more clearly show the advantages of the proposed approach.


3. Effectiveness of the Gaussian-Style Mask: I question the effectiveness of the proposed Gaussian-style mask compared to directly using the coarse mask. Although there is an ablation study in Figure 7 of the Appendix, no experiments address the scenario where the coarse mask is used in the first iteration. Despite potential noise, the coarse mask may still be more informative than a Gaussian-style mask based solely on the distance to a central point, especially for objects with complex structures. The description of “Considering the inaccuracy of coarse mask” is vague, and the author should provide more context on the advantages of the proposed method.  The author is advised to provide more detailed analysis or examples of when and why the Gaussian-style mask is advantageous over the coarse mask.

4. IoU Adaptation: The IoU adaptation strategy seems questionable. When multiple prompts are provided, SAM is trained to predict only one mask via the fourth output token, and the three masks corresponding to the first three output tokens are not utilized. The author proposes a multi-prompt strategy for SAM, but in this context, using the three masks from the single prompt-only scenario seems unnecessary. Additionally, the proposed IoU selection strategy does not show significant improvement (87.1 vs 86.9), as the three predictions converge. I would like the author to clarify why they use the three masks from the single-prompt scenario in their multi-prompt strategy. Additionally,  the author should also provide a more in-depth analysis of the benefits of their IoU adaptation approach given the small improvement observed. For instance, it would be beneficial to examine specific cases where it provides more significant gains.

**Questions:**

See the weakness

---

> ### Author Response · Authors · 2024-11-21
> **Response to Reviewer XLPd (1/3)**
>
> ## Q1:
> > Point Prompts Sampling Method: The distance-based sampling method used is not original, as it is a widely adopted approach for evaluating interactive segmentation models. The author suggests sampling one positive and one negative point, but there is no ablation study to support this design choice (e.g. why not other numbers). Different samples or tasks may have varying characteristics, so the author should provide analysis or empirical ablation to justify this approach.
>
> Although the motivation behind distance-based point prompts is similar to the evaluation used in interactive segmentation, there are notable distinctions between them. Concretely, the evaluation metric **NoC IOU (Number of Clicks)** used in interactive segmentation measures the average number of clicks required to reach a target IOU and is calculated based on the **ground truth mask**. In this setting, each click is placed iteratively at the center of the largest error region between the current prediction and the ground truth mask until the target IOU is achieved or the maximum number of clicks is reached.
>
> In contrast, **our mask refinement setting does not have access to the ground truth mask, making it impossible to iteratively determine multiple points**. For example, while the distance-based principle helps identify the most informative initial point for a given coarse mask, the second distant point is typically the nearest point to the first one, contributing little additional value as a prompt. Consequently, we propose a simpler approach in the mask refinement setting, using one positive and one negative point. This approach is sufficient in our case because we combine these points with box and mask prompts, which offer stronger guidance for SAM.
>
> To further validate this design, we provide an ablation study on DAVIS585, comparing the use of a single positive point versus the combination of positive and negative points. The results in the following **Table R2** demonstrate the effectiveness of our positive-negative design and support its applicability in this setting.
>
> **Table R2: Comparison of single positive point versus the combination of positive and negative points.**
>
> |Point Prompt|mIoU|boundary IoU|
> |---|---|---|
> |Positive|52.5|49.0|
> |Positive+Negative|53.7|49.9|
>
> ---

---

> ### Author Response · Authors · 2024-11-21
> **Response to Reviewer XLPd (2/3)**
>
> ## Q2:
> > Box Prompt Generation Method: Similar concerns apply to the box prompt generation method. A simple baseline would be to take the bounding box over the coarse masks. The appropriateness of the proposed method heavily depends on the quality and characteristics of the coarse mask. While the author provides an ablation study using coarse masks generated by PointWSSIS, this is specific to the dataset and method. The author claims the proposed method is general, but additional analysis or empirical observations are needed to support this claim. The authors are suggested to evaluate the box prompt generation method on a wider range of datasets and coarse mask types to demonstrate its generality. Additionally, proposing a comparison against simpler baselines like the tight bounding box across various datasets would more clearly show the advantages of the proposed approach.
>
> We sincerely appreciate your insightful advice. As we mentioned in Section 3.2, the context-aware elastic box (CEBox) was proposed to address the issue of **incomplete coarse masks**, which are common in instance segmentation predictions.  Following your suggestion, we conducted additional experiments under different settings (e.g., simulated masks on DAVIS585, instance masks generated by NB, and semantic masks from CLIP-ES) for the box prompt and compared the results with the baseline in **Table R3**. The results demonstrate that the context-aware elastic box is highly effective in the instance segmentation setting.  For other settings, the improvement is limited. This is because the strategy specifically targets false-negative pixels in coarse masks, which are rare in these settings. Nonetheless, it shows comparable performance with the baseline.
>
> In practice, coarse masks from different sources usually exhibit varying characteristics and it is challenging to address all scenarios with a single technique. This paper aims to unify several tailed techniques into a universal framework to address different challenges.  Each technique focuses on handling distinct types of noise in coarse masks (e.g., STM for semantic segmentation, CEBox for incomplete coarse mask) and can be flexibly applied based on the source of the mask. Compared to previous methods, which often operate in a closed-world manner, our SAMRefiner framework can flexibly process more scenarios and thus is more versatile and generic.
>
> **Table R3: Results of CEBox under different settings.**
>
> |Methods|Simulated |Instance|Semantic|
> |---|---|---|---|
> |metrics|IoU / boundary IoU|mask AP / boundary AP|mIoU|
> | | | | |
> |baseline|86.9 / 75.1|25.4 / 16.4|78.9|
> |CEBox|87.0 / 75.2|26.1 / 17.0|78.6|

---

> ### Author Response · Authors · 2024-11-21
> **Response to Reviewer XLPd (3/3)**
>
> ## Q3:
> > Effectiveness of the Gaussian-Style Mask: I question the effectiveness of the proposed Gaussian-style mask compared to directly using the coarse mask. Although there is an ablation study in Figure 7 of the Appendix, no experiments address the scenario where the coarse mask is used in the first iteration. Despite potential noise, the coarse mask may still be more informative than a Gaussian-style mask based solely on the distance to a central point, especially for objects with complex structures. The description of “Considering the inaccuracy of coarse mask” is vague, and the author should provide more context on the advantages of the proposed method. The author is advised to provide more detailed analysis or examples of when and why the Gaussian-style mask is advantageous over the coarse mask.
>
> We apologize for the misunderstanding caused by our unclear description. To clarify, we only apply the Gaussian operation to the foreground region of mask, and the Gaussian-style mask is **a generalized form** of the coarse mask. For instance, when the amplitude $\omega$ is set to 1 and the span $\gamma$ is sufficiently large, the Gaussian-style mask is equivalent to the original coarse mask. We provide visualizations of the Gaussian Mask in the Appendix **(Figure 15)** to better explain its functionality. Note that the central point is not the geometry central point of the mask, but the farthest positive point selected by the previous point prompt step.
>
> There are two main reasons for using the Gaussian-style mask:
>
> **1) Compatibility with SAM:** The original SAM doesn't support the binary masks as prompts. This is because the mask prompt merely acts as an auxiliary for point and box in the cascade refinement during SAM pre-training, with the **predicted logits** of the previous iteration as input to guide the next one. Therefore, the mask input for SAM requires logits with **continuous values**, while the original coarse mask is **discrete-valued (0 and1)**. The Gaussian operation can convert the binary mask to continuous, making it compatible to SAM.
>
> **2) The object-centric prior:** The center of an object tends to be positive and feature-discriminative, while uncertainty is mostly located along boundaries. The Gaussian-style mask effectively reduces the weights near boundaries.
>
> As we analyzed in **Section B.5 and Figure 9a**, when $\omega$=1, the performance drops significantly due to the incompatible value space, while the gaussian transformed mask can consistently outperform the original coarse mask under different $\omega$ and $\gamma$.
> We have updated these details in the Appendix and hope this explanation resolves the confusion about the Gaussian-style mask.
>
>
> ---
>
> ## Q4:
> > IoU Adaptation: The IoU adaptation strategy seems questionable. When multiple prompts are provided, SAM is trained to predict only one mask via the fourth output token, and the three masks corresponding to the first three output tokens are not utilized. The author proposes a multi-prompt strategy for SAM, but in this context, using the three masks from the single prompt-only scenario seems unnecessary. Additionally, the proposed IoU selection strategy does not show significant improvement (87.1 vs 86.9), as the three predictions converge. I would like the author to clarify why they use the three masks from the single-prompt scenario in their multi-prompt strategy. Additionally, the author should also provide a more in-depth analysis of the benefits of their IoU adaptation approach given the small improvement observed. For instance, it would be beneficial to examine specific cases where it provides more significant gains.
>
> Although the original SAM uses an individual token when multiple prompts are provided, we empirically observe that selecting the best mask from the remaining three masks based on the IoU prediction yields better performance than the fourth mask, as shown in **Figure 5a**. This is because although the three predictions converge, some details remain different and usually better than the fourth token. We provide visualizations in **Figure 16** to compare the masks generated by different tokens. Though the improvement may not be remarkable in the multi-prompt case, the advantage of IoU adaptation is that **it doesn't require any additional annotated data** and only takes advantage of the priors contained in the target dataset.  SAMRefiner++ serves as a complementary enhancement to SAMRefiner when coarse masks on target datasets can provide high-quality guidance and is not mandatory. We have added the explanation in the Appendix.

---

> > ### Comment · Reviewer_XLPd · 2024-11-21
> > **Response to Q4**
> >
> > I appreciate the inclusion of Figure 16, which provides some visual examples to support the discussion. However, I still have concerns about this aspect. Specifically, it is unclear to me how the output from the first three output tokens could outperform the fourth token in the multi-prompt setting, especially given that the first three tokens are neither utilized nor trained in the multi-prompt scenario. The empirical observations presented are limited, at least in their current form.
> >
> > For instance, in Figure 5(a), the results in the multi-prompt setting show minimal differences (87.1 vs. 86.9), and this is based on a specific dataset and mask type (DAVIS-585). Considering that SAM is a general foundation model and the proposed method is claimed to achieve "universal refinement," stronger evidence is required to support the empirical claims, particularly those that contradict the way SAM was trained.
> >
> > 87.1 vs. 86.9 barely shows any difference on the specific dataset. How would the proposed method work for other datasets from a similar task? How would the proposed method perform on masks involving multi-entity segmentation, such as those in the HQ-SAM training set? Furthermore, how would it handle part masks, such as those in the PACO dataset? Addressing these questions would help validate the universality of the proposed approach.
> >
> > Additionally, while it is acknowledged that the method "does not require any additional annotated data," this alone does not justify introducing a method that exhibits only marginal improvements (87.1 vs. 86.9) on a narrow usage case and deviates from the model's original intended use cases.

---

> ### Comment · Reviewer_XLPd · 2024-11-21
> **Response to Q2:**
>
> Thank you for the explanation. My point is precisely that the type of noise within the mask is highly dependent on several factors: (1) the method used to generate the masks, (2) the type of dataset, and (3) the nature of the segmentation task. I would like to emphasize that the proposed method should explicitly define its intended use cases and acknowledge its limitations.
>
> To illustrate, consider an example where three identical books are placed side by side in an image, and the goal is to refine the instance mask of the middle book. The proposed CEBox method is likely to generate an enlarged box that encompasses all three books, effectively treating them as a single entity. This issue could also extend to part masks and similar scenarios. I would appreciate it if the authors could address this aspect, as it seems to conflict with the claim that the proposed method is capable of performing a "universal mask refinement task

---

> > ### Comment · Reviewer_XLPd · 2024-11-21
> > **Response to Q3**
> >
> > Thanks for the explanation, this makes more sense. I have been using the SAM model to refine the mask previously as well. SAM model is trained by providing the previous coarse mask logits (before binarization), just as the author mentioned. Therefore, not only the value of the coarse mask in SAM training is continuous, but also the value range is different: 0 to 1 (binary masks) vs -100 to 100 (SAM coarse mask). Has the author thought about addressing the value range and scale difference? It does not seem to be so from the equation (3)

---

> > > ### Author Response · Authors · 2024-11-24
> > > **Further response to Q3**
> > >
> > > > Thanks for the explanation, this makes more sense. I have been using the SAM model to refine the mask previously as well. SAM model is trained by providing the previous coarse mask logits (before binarization), just as the author mentioned. Therefore, not only the value of the coarse mask in SAM training is continuous, but also the value range is different: 0 to 1 (binary masks) vs -100 to 100 (SAM coarse mask). Has the author thought about addressing the value range and scale difference? It does not seem to be so from the equation (3).
> > >
> > > We are glad to know that our explanation was helpful in addressing some of your concerns. **Regarding the value range, we think it is exactly the hyperparameter ω we used in equation (3)**, where we scale the value range of the Gaussian map from (0, 1) to (-ω, ω). Specifically, the foreground mask is scaled to (0, ω), while the background value is set to -ω. In **Figure 9**, we provide an ablation study of ω, which shows that a relatively larger ω (greater than 1) is critical for improving the effectiveness of mask prompts, whereas the performance becomes less sensitive when ω is sufficiently large.
> > >
> > > We hope this explanation aligns with your intention and addresses your concerns.

---

> > ### Author Response · Authors · 2024-11-24
> > **Further response to Q2**
> >
> > > Thank you for the explanation. My point is precisely that the type of noise within the mask is highly dependent on several factors: (1) the method used to generate the masks, (2) the type of dataset, and (3) the nature of the segmentation task. I would like to emphasize that the proposed method should explicitly define its intended use cases and acknowledge its limitations.
> > > To illustrate, consider an example where three identical books are placed side by side in an image, and the goal is to refine the instance mask of the middle book. The proposed CEBox method is likely to generate an enlarged box that encompasses all three books, effectively treating them as a single entity. This issue could also extend to part masks and similar scenarios. I would appreciate it if the authors could address this aspect, as it seems to conflict with the claim that the proposed method is capable of performing a "universal mask refinement task.
> >
> > Thank you for your thoughtful suggestion and the provided example. For SAM, the image features of different instances (even within the same category) **exhibit distinct characteristics**. This enables SAM to produce **fine-grained component-level** segments, making it support a variety of downstream applications (e.g., local feature learning as demonstrated in *[1]*).
> >
> > To illustrate this, we analyze feature similarity between different masks in **Figure 19a**. As shown, the features of different instances, even within the same class, display certain differences. This characteristic allows SAM to distinguish between instances effectively (e.g., adjacent books). Similar conclusions can also be drawn for the **part segmentation**, as shown in **Figure 19b**.
> >
> > In our CEBox, a threshold λ is used to determine the necessity to expand the current box in each direction, which is based on the feature similarity in the current box and the surrounding context regions. We believe your concerns can be addressed by flexibly adjusting λ according to different settings. For instance, a relaxed threshold could be applied for general segmentations, while a stricter threshold may be more suitable for fine-grained segmentations, such as distinguishing different instances or components.
> >
> > Additionally, a potential future direction could involve adaptively determining the threshold at the image level. For example, SAM's *"everything mode"* could be used to segment the input image, followed by calculating a feature similarity matrix for these segments. The threshold for each image could then be determined based on this similarity matrix using specific criteria.
> >
> > We sincerely appreciate your valuable advice. We have initially clarified the limitations of our method in **Section C** and have incorporated the discussions of the application scenarios and limitations of our method in the revised PDF **(Section E.7)**.
> >
> > ---
> >
> > *[1] Segment Anything Model is a Good Teacher for Local Feature Learning.*

---

> ### Author Response · Authors · 2024-11-24
> **Further response to Q4 (1/2)**
>
> > I appreciate the inclusion of Figure 16, which provides some visual examples to support the discussion. However, I still have concerns about this aspect. Specifically, it is unclear to me how the output from the first three output tokens could outperform the fourth token in the multi-prompt setting, especially given that the first three tokens are neither utilized nor trained in the multi-prompt scenario. The empirical observations presented are limited, at least in their current form.
>
> > For instance, in Figure 5(a), the results in the multi-prompt setting show minimal differences (87.1 vs. 86.9), and this is based on a specific dataset and mask type (DAVIS-585). Considering that SAM is a general foundation model and the proposed method is claimed to achieve "universal refinement," stronger evidence is required to support the empirical claims, particularly those that contradict the way SAM was trained.
>
> > 87.1 vs. 86.9 barely shows any difference on the specific dataset. How would the proposed method work for other datasets from a similar task? How would the proposed method perform on masks involving multi-entity segmentation, such as those in the HQ-SAM training set? Furthermore, how would it handle part masks, such as those in the PACO dataset? Addressing these questions would help validate the universality of the proposed approach.
>
> > Additionally, while it is acknowledged that the method "does not require any additional annotated data," this alone does not justify introducing a method that exhibits only marginal improvements (87.1 vs. 86.9) on a narrow usage case and deviates from the model's original intended use cases.
>
> Thank you for your thoughtful suggestion. From our experiments, when using multi-prompt, the output of four tokens tends to be similar. As a result, the performance of selecting the best mask from the three masks is comparable to, or slightly better than directly using the fourth token. This common phenomenon **is consistent across different datasets**, as shown in **Table R5**. We hypothesize that the discrepancy between this observation and SAM’s training setup may arise from two factors: 1) The four output tokens are **concatenated** with the prompt tokens, followed by a **self-attention operation** between these tokens. Although the first three tokens are not explicitly supervised by the loss, they participate in the forward pass of the fourth token and may be updated implicitly. 2) During SAM pre-training, a ground-truth mask is iteratively trained for 11 iterations using different sampled prompts (including both single-prompt and multi-prompt), which may encourage the outputs of these two types of tokens to be similar. However, since the official training code has not been released, we regret that a deeper investigation into these details is currently not possible.
>
>
> **Table R5: Results of the mask produced by the fourth token and the remaining three tokens.**
> |Method|DAVIS585|VOC|COCO|COD10K|
> |---|---|---|---|---|
> |4th token|85.9|79.2|26.1|76.4|
> |best mask from first 3 tokens|86.9|79.3|26.1|76.4|
>
>
>
> The underlying rationale of IoU Adaption is that selecting the best mask from a set of similar masks, rather than directly using the fourth token alone, may result in better performance. Ideally, if we use the GT mask to select the best mask, we would get the best performance (upper limit). As shown in **Table R6**, the upper limit tends to be much higher than both using the fourth token and SAM's IoU based mask selection (e.g., 90.4 vs 86.9), suggesting room for improvement in SAM's IoU prediction on target datasets. In the mask refinement task, GT masks are not available. As an alternative, we use coarse masks, which impose strict requirements on the quality of these coarse masks. Although the IoU Adaption method is dataset-dependent and may not achieve remarkable results on all datasets, it offers a potential solution that does not rely on additional ground-truth annotations and provides valuable insight for minimal adaption of SAM (e.g., promoting recognition of the critical role of the IoU head). We have updated **Figure 5a** by including the GT IoU based performance to make it clearer to understand.
>
> **Table R6: Comparisons of selecting the best mask using different IoU.**
> |Method|DAVIS585|VOC|
> |---|---|---|
> |coarse mask|81.4|70.8|
> |4th token|85.9|79.2|
> |best mask from 3 tokens (SAM's IoU predcition)|86.9|79.3|
> |best mask from 3 tokens (IoU Adaption)|87.1|80.1|
> |best mask from 3 tokens (GT IoU)|90.4|81.6|

---

> ### Author Response · Authors · 2024-11-24
> **Further response to Q4 (2/2)**
>
> > I appreciate the inclusion of Figure 16, which provides some visual examples to support the discussion. However, I still have concerns about this aspect. Specifically, it is unclear to me how the output from the first three output tokens could outperform the fourth token in the multi-prompt setting, especially given that the first three tokens are neither utilized nor trained in the multi-prompt scenario. The empirical observations presented are limited, at least in their current form.
>
> > For instance, in Figure 5(a), the results in the multi-prompt setting show minimal differences (87.1 vs. 86.9), and this is based on a specific dataset and mask type (DAVIS-585). Considering that SAM is a general foundation model and the proposed method is claimed to achieve "universal refinement," stronger evidence is required to support the empirical claims, particularly those that contradict the way SAM was trained.
>
> > 87.1 vs. 86.9 barely shows any difference on the specific dataset. How would the proposed method work for other datasets from a similar task? How would the proposed method perform on masks involving multi-entity segmentation, such as those in the HQ-SAM training set? Furthermore, how would it handle part masks, such as those in the PACO dataset? Addressing these questions would help validate the universality of the proposed approach.
>
> > Additionally, while it is acknowledged that the method "does not require any additional annotated data," this alone does not justify introducing a method that exhibits only marginal improvements (87.1 vs. 86.9) on a narrow usage case and deviates from the model's original intended use cases.
>
> We would like to emphasize the distinction between SAMRefiner and SAMRefiner++. SAMRefiner is a **training-free** method that refines masks using noise-tolerant prompts. **It retains most of the characteristics of the original SAM and inherits its "universal capability."** In contrast, SAMRefiner++ refers to **the combination of SAMRefiner and IoU Adaption**, which require additional training on target datasets. This method is specifically tailored for certain conditions and **has strict prerequisites**, such as the quality of coarse masks, which is **dataset-dependent**. As a result, SAMRefiner++ is not intended to be a universal method. Instead, it offers a potential approach to achieve further improvements without requiring additional annotations under certain conditions (e.g., 0.2% on DAVIS585 and 0.8% on VOC). That is why we separate the IoU Adaptation step from SAMRefiner and refer to the extended method as SAMRefiner++.  We acknowledge the limitations of SAMRefiner++ and have discussed them in detail in the revised PDF (Section E.7).
>
> Actually, SAMRefiner can perform well in most cases. To validate the generality of SAMRefiner, we followed your suggestion and conducted experiments under various settings. In **Table R7**, we report results on the part segmentation dataset PACO. Coarse masks were generated using MaskRCNN, and we measured the mask AP for part objects before and after refinement. The results demonstrate that SAMRefiner is able to improve mask quality in this setting, aligning with the analysis presented earlier.
>
> **Table R7: Results on part segmentation dataset PACO.**
> |Method|PACO|
> |---|---|
> |Coarse Mask|18.6|
> |SAMRefiner|19.3|
>
> Additionally, in **Table R8**, we evaluate SAMRefiner on the fine-grained dataset BIG (used in HQ-SAM) and the challenging concealed object dataset COD10K. Coarse masks were generated using existing segmentation models and then refined using SAMRefiner. The results indicate that SAMRefiner can effectively enhance the quality of coarse masks in both datasets. Moreover, SAMRefiner can be seamlessly integrated into other SAM variants, such as HQ-SAM. Leveraging this more powerful model further improves the performance, as shown in our experiments (**Section B.3**).
>
> **Table R8: Results on the BIG and COD10K (IoU / boundary IoU).**
> |Method|BIG|COD10K|
> |---|---|---|
> |Coarse Mask|89.4 / 60.2|73.8 / 55.8|
> |SAMRefiner|92.2 / 72.1|76.4 / 60.7|
> |SAMRefiner (HQ-SAM)|93.9 / 74.8|77.1 / 61.9|
>
> We hope we have correctly understood your concerns. If there are any remaining questions or concerns, please feel free to reach out for further discussion.

---

> > ### Comment · Reviewer_XLPd · 2024-11-24
> > **Thanks for the detailed explanation**
> >
> > I appreciate the author's response. The explanation for Q2 and Q3 is clear and addresses my concerns. For Q4, the quantitative result resolves some of my concerns. However, considering the limited performance gain of the IoU adaptation, I still have reservations about that.  In general, I thank the author for their effort in addressing most of my concerns. I am adjusting the rating to 6 to reflect that.

---

> > > ### Author Response · Authors · 2024-11-25
> > >
> > > We deeply appreciate your acknowledgment of our work and the responses we provided. Best wishes to you.

---

### Official Review · Reviewer_NC8S · 2024-11-03

**Soundness:** 3
**Presentation:** 2
**Contribution:** 3
**Rating:** 6
**Confidence:** 4

**Summary:**

The paper proposes SAMRefiner, a novel framework for mask refinement using the Segment Anything Model (SAM). It contains several techniques: noise-tolerant prompting scheme, split-then-merge strategy, and IoU adaption. By combining these techniques, the proposed SAMRefiner improves the quality of the SAM-generated masks.

**Strengths:**

1. The proposed SAMRefiner offers a unique solution to the mask refinement problem, which is an important yet under-explored area. By adapting SAM for this task, the authors address a practical issue of improving the quality of pre-existing coarse masks, which can have significant implications for reducing annotation costs and enhancing the performance of downstream segmentation models.

2. The introduction of the noise-tolerant prompting scheme, including the multi-prompt excavation strategy, is interesting.

3. The split-then-merge (STM) is a practical solution to handle the challenges of multiple objects in the mask.

4. The IoU adaption step is another interesting technique. By introducing a self-boosted method to enhance SAM's IoU prediction ability on specific datasets using coarse mask priors, the authors demonstrate a way to further improve the performance of the framework without additional annotation.

5. The authors conduct a comprehensive set of experiments on various benchmarks, including DAVIS-585, COCO, and PASCAL VOC, under different settings (such as instance and semantic segmentation with incomplete supervision). This wide range of evaluations helps to demonstrate the versatility and effectiveness of SAMRefiner in different scenarios.

6. The comparison with state-of-the-art model-agnostic refinement methods is thorough. The detailed performance metrics presented (e.g., IoU, mask AP, mIoU) and the analysis of the results provide strong evidence of the superiority of SAMRefiner in many cases, especially in terms of its robustness to mask noise and its ability to improve performance across diverse datasets.

**Weaknesses:**

1. The explanation of how the proposed method overcomes the limitations of SAM in handling multi-object cases, especially in the context of semantic segmentation, could be more detailed. While the STM pipeline is introduced, its interaction with the overall framework and the specific advantages it brings compared to other possible approaches are not entirely clear. In addition, I suggest the authors to use a flowchart and visualize some examples after each stage in STM.

2. Although the paper compares with several state-of-the-art refinement methods, it could provide a more in-depth analysis of why some methods perform better or worse in different scenarios. For example, a more detailed discussion of the differences between SAMRefiner and methods like CascadePSP and CRM in terms of their design principles and how these differences lead to performance variations would be beneficial. It is better to compare the key design principles of SAMRefiner with others in a table or a figure.

3. Some implementation details are missing or could be further elaborated. For instance, the specific hyperparameters used in different experiments (beyond the ones mentioned for the IoU adaption step) are not always clearly stated. This makes it difficult for other researchers to fully reproduce the experiments and evaluate the method under different conditions. The authors are suggested to include these implementations and put them in the appendix.

4. The paper would benefit from more visualizations to support the understanding of the proposed method and the experimental results. For example, visual comparisons of the mask refinement process using different prompts and techniques could provide a better understanding of how the method works and why it is effective. Additionally, visualizations of the failure cases and their analysis could help to identify the limitations of the method more clearly.

**Questions:**

See "Weakness"

---

> ### Author Response · Authors · 2024-11-21
> **Response to Reviewer NC8S (1/2)**
>
> ## Q1:
> > The explanation of how the proposed method overcomes the limitations of SAM in handling multi-object cases, especially in the context of semantic segmentation, could be more detailed. While the STM pipeline is introduced, its interaction with the overall framework and the specific advantages it brings compared to other possible approaches are not entirely clear. In addition, I suggest the authors to use a flowchart and visualize some examples after each stage in STM.
>
> Thanks for your suggestion. The underlying rationale of STM is to convert semantic masks with multiple objects into instance masks to ensure better compatibility with SAM. This step is performed before using SAMRefiner to extract prompts. While we had initially provided a pseudo-code for STM in the Appendix **(Algorithm 1)**, we have now followed your suggestion and added a flowchart with visualizations for each stage in STM **(Figure 13)**. Furthermore, we show the advantages of STM by visually comparing it with the baseline (without STM) in **Figure 14**. These additional contents are presented in the revised PDF **(Page 22 in the Appendix)**. We hope these revisions offer a clearer explanation of our STM strategy.
>
> ---
>
> ## Q2:
> > Although the paper compares with several state-of-the-art refinement methods, it could provide a more in-depth analysis of why some methods perform better or worse in different scenarios. For example, a more detailed discussion of the differences between SAMRefiner and methods like CascadePSP and CRM in terms of their design principles and how these differences lead to performance variations would be beneficial. It is better to compare the key design principles of SAMRefiner with others in a table or a figure.
>
>
> **Table R1: Comparison of different mask refinement methods.**
>
>
> | Method       | Design Principle                                                             | Architecture | Training Data                                          | Advantages                                   | Drawbacks                                    |
> |--------------|------------------------------------------------------------------------------|--------------|-------------------------------------------------------|---------------------------------------------|---------------------------------------------|
> | **dense CRF**    | Maximize label agreement between pixels with similar low-level color         | None         | None                                                  | Training-free, Easy to use            | Inaccurate                                |
> | **CascadePSP**   | Align the feature map with the refinement target in a cascade fashion       | CNN          | MSRA-10K, DUT-OMRON, ECSSD, FSS-1000    |  Class-Agnostic, Accurate on semantic segmentation | Task-dependent, Inefficient         |
> | **CRM**          | Align the feature map with the refinement target continuously               | CNN          | MSRA-10K, DUT-OMRON, ECSSD, FSS-1000    | Class-Agnostic, Accurate on semantic segmentation | Task-dependent, Inefficient           |
> | **SAMrefiner**   | Design noise-tolerance prompts to enable SAM for mask refinement             | Transformer  | SA-1B                                                 | Class-agnostic, Task-agnostic, Accurate, Efficient | Objects with intricate architecture        |
>
>
> We provide a detailed discussion of the differences between SAMRefiner and related methods (dense CRF, CascadePSP, CRM) in terms of the design principle, architecture, training data, advantages, and drawbacks in **Table R1**. Among these methods, dense CRF is a training-free post-process approach based on low-level color characteristics, making it efficient and easy to use. However, it struggles in complex scenarios due to its lack of high-level semantic context. CascadePSP and CRM, on the other hand, focus on aligning the feature map with the refinement target using CNN-based architectures. They are trained on a combined dataset with extremely accurate mask annotations and demonstrate strong performance on semantic segmentation tasks. Nevertheless, their performance on instance segmentation is less competitive, primarily due to the absence of complex cases in their training data and the inherent limitations of CNNs. Additionally, the cascade structure of CascadePSP and the multi-resolution inference required by CRM make them inefficient when handling masks with a large number of objects.
>
> In contrast, SAMRefiner leverages the strengths of SAM by designing noise-tolerant prompts specifically for mask refinement tasks. This approach achieves better accuracy and efficiency compared to existing methods. Nevertheless, it may underperform for objects with intricate structures, a limitation inherited from SAM itself. This issue can be addressed using enhanced variants, such as HQ-SAM[1],  as the experiments we conducted in **Appendix Section B.3**.
>
> ---
> [1] Segment anything in high quality.

---

> ### Author Response · Authors · 2024-11-21
> **Response to Reviewer NC8S (2/2)**
>
> ## Q3:
> > Some implementation details are missing or could be further elaborated. For instance, the specific hyperparameters used in different experiments (beyond the ones mentioned for the IoU adaption step) are not always clearly stated. This makes it difficult for other researchers to fully reproduce the experiments and evaluate the method under different conditions. The authors are suggested to include these implementations and put them in the appendix.
>
> Thanks for your advice. For instance segmentation, the threshold $\lambda$ is set to 0.1 for the box prompt. For semantic segmentation, $\mu$ used in STM is set to 0.5. The factors $\omega, \gamma$ for Gaussian distribution are set to 15 and 4 by default. We have updated these implementations **in the Appendix (Section A.2)**. We also conducted ablation studies about these hyperparameters **in the Appendix (Section B.5)**. We will release our codes and configurations to support reproducibility.
>
> ---
>
> ## Q4:
> > The paper would benefit from more visualizations to support the understanding of the proposed method and the experimental results. For example, visual comparisons of the mask refinement process using different prompts and techniques could provide a better understanding of how the method works and why it is effective. Additionally, visualizations of the failure cases and their analysis could help to identify the limitations of the method more clearly.
>
> Thanks for your thoughtful suggestion. We have included several visualizations in the Appendix to aid understanding. For example, **in Figure 10**, we analyze some failure cases on semantic segmentation. **In Figure 11-12**, we provided more visualizations of SAMRefiner on different datasets and compared it with other techniques. We hope these examples help clarify our work. If there are specific areas of interest, we would be happy to provide further visualizations upon request.

---

> > ### Comment · Reviewer_NC8S · 2024-11-24
> > **Post-Rebuttal**
> >
> > Thanks for the response from the authors. I have carefully checked the response from the authors, and most of my concerns have been addressed. I am happy to maintain my initial score.

---

> > > ### Author Response · Authors · 2024-11-24
> > >
> > > We deeply appreciate your acknowledgment of our work and the responses we provided. Best wishes to you.

---

### Author Response · Authors · 2024-11-21
**General Response**

We sincerely appreciate all reviewers for their valuable feedback and efforts in evaluating our work, which have greatly helped us improve the quality of our paper. We also deeply appreciate the substantial efforts invested by all the ACs and PCs throughout the review process.

To address the concerns raised by the reviewers, we have provided detailed responses to each reviewer individually. The revised version of the PDF includes additional figures and detailed explanations, with all updates **highlighted in blue** for ease of reference. Please feel free to consult the updated paper, and we are happy to offer further clarification if needed.

---

## Update

We sincerely appreciate all reviewers’ quick response and active participation in the discussion, which have significantly contributed to improving the quality and clarity of our manuscript.  We are pleased that the reviewers have recognized the highlights of our work, such as offering a unique and practical solution for the meaningful mask refinement task **(Reviewer NC8S, XLPd, a711)**, the novel and effective designs of several noise-tolerant prompts **(all reviewers)**, the interesting self-boosted IoU adaption technique **(Reviewer NC8S, a711)**, the clear writing **(Reviewer a711, ukvm)**, as well as the thorough experiments to demonstrate the versatility and effectiveness of SAMRefiner **(Reviewer NC8S, ukvm)**. During the discussion, we addressed the reviewers’ concerns by clarifying technical details and limitations, as well as providing additional comparative experiments and visualizations. We are pleased that these responses were helpful in resolving the reviewers’ concerns and that all reviewers expressed a positive perspective on the paper after the discussion.

Broadly Speaking, our SAMRefiner is intended to enhance the quality of pre-existing coarse masks, helping to reduce annotation costs and improve the performance of downstream segmentation models. We tame SAM for universal mask refinement by designing several noise-tolerant prompts, including distance-guided points, context-aware elastic bounding boxes, and Gaussian-style masks. A split-then-merge (STM) pipeline is introduced to handle the multi-object case in semantic segmentation. SAMRefiner is training-free and retains the universal characteristics of the original SAM, remaining agnostic to categories, models, and tasks. We further explore a dataset-specific variant, SAMRefiner++, by incorporating SAMRefiner with the proposed IoU Adaption. While the improvement is data-dependent, this self-boosted strategy could enhance SAM's IoU prediction ability on specific datasets using coarse mask priors without additional annotation (Reviewer NC8S, a711), offering novel insights for minimal adaption of SAM. Experiments across a wide range of benchmarks validate the effectiveness and generality of the proposed method in different scenarios. We believe our investigations and observations on SAM could offer new perspectives for the community, and SAMRefiner shows great potential for advancing mask refinement and acting as a versatile post-processing tool.

Thanks again to all the reviewers for their insightful feedback and valuable efforts in evaluating our work, as well as to the ACs and PCs for their dedicated contributions throughout the review process.

Best regards,

Authors

---

### Meta-Review · Area_Chair_4Qve · 2024-12-21

**Metareview:**

The paper introduces SAMRefiner, a framework for refining coarse segmentation masks using the Segment Anything Model (SAM). The reviewers found that this paper tackles practical challenges in segmentation by improving coarse masks, reducing annotation costs, novel prompt designs and task-agnostic adaptability make SAMRefiner versatile, comprehensive evaluations demonstrate strong performance across datasets. Some concerns are raised by the reviewers: IoU adaptation shows limited improvement (e.g., 87.1 vs. 86.9 on DAVIS-585), performance heavily relies on the initial mask’s quality, lacking details on hyperparameters and reproducibility. Given its technical contributions, practical relevance, and good experimental results, SAMRefiner is a valuable addition to segmentation literature. However, the novelty of specific techniques and marginal gains from IoU adaptation warrant consideration. Overall, the paper is recommended for acceptance with the expectation that future iterations address the highlighted concerns.

**Additional Comments On Reviewer Discussion:**

The paper received four 6 scores in the final rating. Reviewer XLPd raised rating to 6 after the raised concerns regarding Box Prompt Generation Method and Effectiveness of the Gaussian-Style Mask were addressed by the authors in rebuttal. Reviewer ukvm and Reviewer NC8S replied to keep the initial rating 6 after reading the authors' rebuttal. There is a consensus that the paper is marginally above the acceptance threshold.

---

### Decision · Program_Chairs · 2025-01-22

Accept (Poster)